# Identifying Treatment Effects under Unobserved Confounding by Causal Representation Learning

## Abstract

As an important problem of causal inference, we discuss the estimation of treatment effects under the existence of unobserved confounding. By representing the confounder as a latent variable, we propose Counterfactual VAE, a new variant of variational autoencoder, based on recent advances in identifiability of representation learning. Combining the identifiability and classical identification results of causal inference, under mild assumptions on the generative model and with small noise on the outcome, we theoretically show that the confounder is identifiable up to an affine transformation and then the treatment effects can be identified. Experiments on synthetic and semi-synthetic datasets demonstrate that our method matches the state-of-the-art, even under settings violating our formal assumptions.

## 1 Introduction

Causal inference (Imbens & Rubin, 2015; Pearl, 2009), i.e, estimating causal effects of interventions, is a fundamental problem across many domains. In this work, we focus on the estimation of treatment effects, e.g., effects of public policies or a new drug, based on a set of observations consisting of binary labels for treatment / control (non-treated), outcome, and other covariates. The fundamental difficulty of causal inference is that we never have observations of *counterfactual* outcomes, which would have been if we had made another decision (treatment or control). While the ideal protocol for causal inference is randomized controlled trials (RCTs), they often have ethical and practical issues, or are prohibitively expensive. Thus, causal inference from observational data is indispensable, though they introduce other challenges. Perhaps the most crucial one is *confounding*: there might be variables (called *confounders*) that causally affect both the treatment and the outcome, and spurious correlation follows.

Most of works in causal inference rely on the *unconfoundedness* assumption that appropriate covariates are collected so that the confounding can be controlled by conditioning on or *adjusting* for those variables. This is still challenging, due to systematic difference of the distributions of the covariates between the treatment and control groups. One classical way of dealing with this difference is re-weighting (Horvitz & Thompson, 1952). There are semi-parametric methods, which have better finite sample performance, e.g. TMLE (Van der Laan & Rose, 2011), and also non-parametric, tree-based, methods, e.g. Causal Forests (CF) (Wager & Athey, 2018). Notably, there is a recent rise of interest in *representation learning* for causal inference starting from Johansson et al. (2016).

There are a few lines of works that challenge the difficult but important problem of causal inference under *unobserved confounding*. Without covariates we can adjust for, many of them assume special structures among the variables, such as instrumental variables (IVs) (Angrist et al., 1996), proxy variables (Miao et al., 2018), network structure (Ogburn, 2018), and multiple causes (Wang & Blei, 2019). Among them, instrumental variables and proxy (or surrogate) variables are most commonly exploited. *Instrumental variables* are not affected by unobserved confounders, influencing the outcome only through the treatment. On the other hand, *proxy variables* are causally connected to unobserved confounders, but are not confounding the treatment and outcome by themselves. Other methods use restrictive parametric models (Allman et al., 2009), or only give interval estimation (Manski, 2009; Kallus et al., 2019).

In this work, we address the problem of estimating treatment effects under unobserved confounding. We further discuss the *individual-level* treatment effect, which measures the treatment effect conditioned on the covariate, for example, on a patient's personal data. To model the problem, we regard the covariate as a proxy variable and the confounder as a latent variable in representation learning.

Our method particularly exploits the recent advance of *identifiability* of representation learning for VAE (Khemakhem et al., 2020). The hallmark of deep neural networks (NNs) might be that they can learn representations of data. It is desirable that the learned representations are *interpretable*, that is, in approximately the same relationship to the latent sources for each down-stream task. A principled approach to this is identifiability, that is, when optimizing our learning objective w.r.t. the representation function, only a unique optimum will be returned. Our method builds on this and further provides the stronger identifiability of representations that is needed in causal inference.

The proposed method is also based firmly on the well-established results in causal inference. In many works exploiting proxies, it is assumed that the proxies are independent of the outcome given the confounder (Greenland, 1980; Rothman et al., 2008; Kuroki & Pearl, 2014). This also motivates our method. Further, our method naturally combines a new VAE architecture with the classical result of Rosenbaum & Rubin (1983) regarding the sufficient information for identification of treatment effects, showing identifiability proof of both latent representations and treatment effects.

The main **contributions** of this paper are as follows: 1) interpretable, causal representation learning by a new VAE architecture for estimating treatment effects under unobserved confounding; 2) theoretical analysis of the identifiability of representation and treatment effect; 3) experimental study on diverse settings showing performance of state-of-the-art.

## 2    RELATED WORK

**Identifiability of representation learning.** With recent advances in nonlinear ICA, identifiability of representations is proved under a number of settings, e.g., auxiliary task for representation learning (Hyvärinen & Morioka, 2016; Hyvärinen et al., 2019) and VAE (Khemakhem et al., 2020). Recently, Roeder et al. (2020) extends the the result to include a wide class of state-of-the-art deep discriminative models. The results are exploited in bivariate causal discovery (Wu & Fukumizu, 2020) and structure learning (Yang et al., 2020). To the best of our knowledge, this work is the first to explore this new possibility in causal inference.

**Representation learning for causal inference.** Recently, researchers start to design representation learning methods for causal inference, but mostly limited to unconfounded settings. Some methods focus on learning a balanced covariate representation, e.g., BLR/BNN (Johansson et al., 2016), and TARnet/CFR (Shalit et al., 2017). Adding to this, Yao et al. (2018) also exploits the local similarity of between data points. Shi et al. (2019) uses similar architecture to TARnet, considering the importance of treatment probability. There are also methods using GAN (Yoon et al., 2018, GANITE) and Gaussian process (Alaa & van der Schaar, 2017). Our method adds to these by also tackling the harder problem of unobserved confounding.

**Causal inference with auxiliary structures.** Both our method and CEVAE (Louizos et al., 2017) are motivated by exploiting proxies and use VAE as a learning method. However, CEVAE assumes a specific causal graph where the covariates should be independent of the treatment given the confounder. Further, CEVAE relies on the assumption that VAE can recover the true latent distribution. Kallus et al. (2018) uses matrix factorization to infer the confounders from proxy variables, and gives consistent ATE estimator and its error bound. Miao et al. (2018) established conditions for identification using more general proxies, but without practical estimation method. Note that, two active lines of works in machine learning exist in their own right, exploiting IV (Hartford et al., 2017) and network structure (Veitch et al., 2019).

## 3    SETUP AND PRELIMINARIES

### 3.1    TREATMENT EFFECTS AND CONFOUNDERS

Following Imbens & Rubin (2015), we begin by introducing *potential outcomes* (or *counterfactual outcomes*) $\mathrm{y}(t), t = 0, 1$. $\mathrm{y}(t)$ is the outcome we *would* observe, if we applied treatment value $t$.

Note that, for a unit under research, we can observe only one of $y(0)$ or $y(1)$, corresponding to which factual treatment we have applied. This is the *fundamental problem of causal inference*.

We write expected potential outcomes, conditioned on covariate(s) $\mathbf{x} = \boldsymbol{x}$ as $\mu_t(\boldsymbol{x}) = \mathbb{E}(\mathrm{y}(t)|\mathbf{x} = \boldsymbol{x})$. The estimands in this work are the causal effects, which are Conditional Average Treatment Effect (CATE) and Average Treatment Effect (ATE) defined by

$$\tau(\boldsymbol{x}) = \mu_1(\boldsymbol{x}) - \mu_0(\boldsymbol{x}), \qquad ATE = \mathbb{E}(\tau(\boldsymbol{x})) \tag{1}$$

CATE can be understood as an *individual-level* treatment effect, if conditioned on high dimensional and highly diverse covariates.

In general, we need three assumptions for identification (Rubin, 2005). There should exist variable $\mathbf{z} \in \mathbb{R}^n$ satisfies *ignorability* $(\mathrm{y}(0), \mathrm{y}(1) \perp\!\!\!\perp \mathrm{t}|\mathbf{z})$ and *positivity* $(\forall \boldsymbol{z}, t : p(\mathrm{t} = t|\mathbf{z} = \boldsymbol{z}) > 0)$, and also given the *consistency* of counterfactuals $(\mathrm{y} = \mathrm{y}(t)$ if $\mathrm{t} = t)$ (See Appendix for explanations). Then, treatment effects can be *identified* by:

$$\mu_t(\boldsymbol{x}) = \mathbb{E}(\mathbb{E}(\mathrm{y}(t)|\mathbf{z}, \mathbf{x} = \boldsymbol{x})) = \mathbb{E}(\mathbb{E}(\mathrm{y}|\mathbf{z}, \mathbf{x} = \boldsymbol{x}, \mathrm{t} = t)) = \int(\int p(y|\boldsymbol{z}, \boldsymbol{x}, t)y dy)p(\boldsymbol{z}|\boldsymbol{x})d\boldsymbol{z} \tag{2}$$

The second equality uses the three conditions. We say that *strong ignorability* holds when we have both ignorability and positivity. In this work, we consider *unobserved confounding*, that is, we assume the existence of *confounder(s)* $\mathbf{z}$, satisfying the three conditions, but it is (partially)[1] unobserved.

The following theorem adapted from Rosenbaum & Rubin (1983) is central to causal inference and we will use it for motivating and justifying our method. Such function $\boldsymbol{b}(\mathbf{z})$ is called a *balancing score* (of $\mathbf{z}$). Obviously, the *propensity score* $e(\mathbf{z}) := p(\mathrm{t} = 1|\mathbf{z})$, the propensity of assigning the treatment given $\mathbf{z}$, is a balancing score (with $f$ be the identity function).

**Theorem 1** (Balancing score). *Let $\boldsymbol{b}(\mathbf{z})$ be a function of random variable $\mathbf{z}$. Then $\mathrm{t} \perp\!\!\!\perp \mathbf{z}|\boldsymbol{b}(\mathbf{z})$ if and only if $f(\boldsymbol{b}(\mathbf{z})) = p(\mathrm{t} = 1|\mathbf{z}) := e(\mathbf{z})$ for some function $f$ (or more formally, $e(\mathbf{z})$ is $\boldsymbol{b}(\mathbf{z})$-measurable). Assume further that $\mathbf{z}$ satisfies strong ignorability, then so does $\boldsymbol{b}(\mathbf{z})$.*

## 3.2 VARIATIONAL AUTOENCODERS

Variational autoencoders (VAEs) (Kingma et al., 2019) are a class of latent variable models with latent variable $\mathbf{z}$, and observed variable $\mathbf{y}$ is generated by the decoder $p_{\boldsymbol{\theta}}(\mathbf{y}|\mathbf{z})$. The variational lower bound of the log-likelihood is written as:

$$\log p(\mathbf{y}) \geq \log p(\mathbf{y}) - D_{\mathrm{KL}}(q(\mathbf{z}|\mathbf{y})\|p(\mathbf{z}|\mathbf{y})) = \underbrace{\mathbb{E}_{\mathbf{z} \sim q} \log p_{\boldsymbol{\theta}}(\mathbf{y}|\mathbf{z}) - D_{\mathrm{KL}}(q_{\boldsymbol{\phi}}(\mathbf{z}|\mathbf{y})\|p(\mathbf{z}))}_{\mathcal{L}_{VAE}(\mathbf{y};\boldsymbol{\theta},\boldsymbol{\phi})}, \tag{3}$$

where the encoder $q_{\boldsymbol{\phi}}(\mathbf{z}|\mathbf{y})$ is introduced to approximate the true posterior $p(\mathbf{z}|\mathbf{y})$ and $D_{\mathrm{KL}}$ denotes KL divergence. The decoder $p_{\boldsymbol{\theta}}$ and encoder $q_{\boldsymbol{\phi}}$ are usually parametrized by NNs. We will omit the parameters $\boldsymbol{\theta}, \boldsymbol{\phi}$ in notations when appropriate. Using the reparameterization trick (Kingma & Welling, 2014) and optimizing the evidence lower bound (ELBO) $\mathbb{E}_{\mathbf{y} \sim \mathcal{D}}(\mathcal{L}(\boldsymbol{y}))$ with data $\mathcal{D}$, we train the VAE efficiently. Conditional VAE (CVAE) adds a conditioning variable $\mathbf{c}$ to (3) (See Appendix for details).

As mentioned, identifiable VAE (iVAE) (Khemakhem et al., 2020) provides the first identifiability result for VAE, using auxiliary variable $\mathbf{u}$. It assumes $\mathbf{y} \perp\!\!\!\perp \mathbf{u}|\mathbf{z}$, that is, $p(\mathbf{y}|\mathbf{z}, \mathbf{u}) = p(\mathbf{y}|\mathbf{z})$. The variational lower bound is

$$\log p(\mathbf{y}|\mathbf{u}) \geq \underbrace{\mathbb{E}_{\mathbf{z} \sim q} \log p_{\boldsymbol{f}}(\mathbf{y}|\mathbf{z}) - D_{\mathrm{KL}}(q(\mathbf{z}|\mathbf{y}, \mathbf{u})\|p_{\boldsymbol{T},\boldsymbol{\lambda}}(\mathbf{z}|\mathbf{u}))}_{\mathcal{L}_{iVAE}(\mathbf{y},\mathbf{u})} \tag{4}$$

where $\mathbf{y} = \boldsymbol{f}(\mathbf{z}) + \boldsymbol{\epsilon}$, $\boldsymbol{\epsilon}$ is additive noise and $\mathbf{z}$ has exponential family distribution with sufficient statistics $\boldsymbol{T}$ and parameter $\boldsymbol{\lambda}(\mathbf{u})$. Note that, unlike CVAE, the decoder does *not* depend on $\mathbf{u}$ due to the independence assumption.

Here identifiability means that the functional parameters $(\boldsymbol{f}, \boldsymbol{T}, \boldsymbol{\lambda})$ can be identified (learned) up to a simple transformation.

## 4 VAE ARCHITECTURE FOR CAUSAL REPRESENTATION LEARNING

---

[1] This allows the existence of *observed* confounders in $\mathbf{x}$. As we will see, since $\mathbf{z}$ is the latent variable(s) for VAE and is learned from covariates $\mathbf{x}$ by the VAE, it can contain *all* confounders in principle. Our method will extract the confounding part of $\mathbf{x}$ into $\mathbf{z}$.

The probabilistic model of our VAE follows naturally from the right most side of (2), where the involved distribution is $p(\mathbf{y}, \mathbf{z}|\mathbf{x}, \mathbf{t}) = p(\mathbf{y}|\mathbf{z}, \mathbf{x}, \mathbf{t})p(\mathbf{z}|\mathbf{x}, \mathbf{t})$. If we treat covariate $\mathbf{x}$ as a proxy and assume further the conditional independence $\mathbf{y} \perp\!\!\!\perp \mathbf{x}|\mathbf{z}, \mathbf{t}$ as in most work exploiting proxies, we have

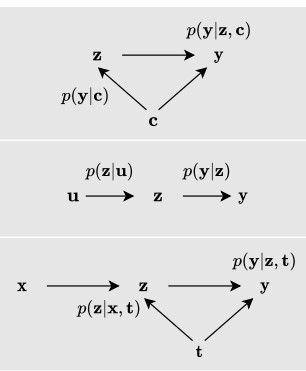

Figure 1: Graphical models of the decoders. From top: CVAE, iVAE, and CFVAE. The encoders for the VAEs are similar: they take all observed variables and build approximate posteriors.

$$p(\mathbf{y}, \mathbf{z}|\mathbf{x}, \mathbf{t}) = p(\mathbf{y}|\mathbf{z}, \mathbf{t})p(\mathbf{z}|\mathbf{x}, \mathbf{t}) \qquad (5)$$

A natural next step is to design a VAE for this joint distribution, which learns to recover a causal representation of $\mathbf{z}$. By "causal", we mean the representation can be used to identify or estimate treatment effects. Recovering the true confounder $\mathbf{z}$ would be great, but this is not required, as shown in Theorem 1, which says, for identification of treatment effects, we only need to have $\boldsymbol{b}(\mathbf{z})$, a causal representation of $\mathbf{z}$, which contains the information of propensity score $e(\mathbf{z})$, the part of $\mathbf{z}$ that is relevant to treatment assignment.

We are a step away from the VAE architecture now. Note that (5) has similar factorization with iVAE: $p(\mathbf{y}, \mathbf{z}|\mathbf{u}) = p(\mathbf{y}|\mathbf{z})p(\mathbf{z}|\mathbf{u})$ from $\mathbf{y} \perp\!\!\!\perp \mathbf{u}|\mathbf{z}$, meaning we can use our covariate $\mathbf{x}$ as auxiliary variable $\mathbf{u}$ in iVAE from the independence assumption of proxy variable. Further from the conditioning on $\mathbf{t}$ in (5), we design a VAE architecture as a combination of CVAE and iVAE, with treatment $\mathbf{t}$ and covariate $\mathbf{x}$ as conditioning and auxiliary variable, respectively. The ELBO can be derived as

$$
\begin{aligned}
\log p(\mathbf{y}|\mathbf{x}, \mathbf{t}) &\geq \log p(\mathbf{y}|\mathbf{x}, \mathbf{t}) - D_{\mathrm{KL}}(q(\mathbf{z}|\mathbf{x}, \mathbf{y}, \mathbf{t}) \| p(\mathbf{z}|\mathbf{x}, \mathbf{y}, \mathbf{t})) \\
&= \mathbb{E}_{\mathbf{z}\sim q} \log p(\mathbf{y}|\mathbf{z}, \mathbf{t}) - D_{\mathrm{KL}}(q(\mathbf{z}|\mathbf{x}, \mathbf{y}, \mathbf{t}) \| p(\mathbf{z}|\mathbf{x}, \mathbf{t})) := \mathcal{L}_{CFVAE}(\mathbf{x}, \mathbf{y}, \mathbf{t}).
\end{aligned}
\qquad (6)
$$

As in iVAE, the decoder drops the dependence on $\mathbf{x}$. We name this architecture the *Counterfactual VAE* (CFVAE). Figure 1 depicts the relationship of CVAE, iVAE, and CFVAE.

We detail parameterization of CFVAE. The decoder $p_{\boldsymbol{f}, \boldsymbol{g}}(\mathbf{y}|\mathbf{z}, \mathbf{t})$, conditional prior $p_{\boldsymbol{h}, \boldsymbol{k}}(\mathbf{z}|\mathbf{x}, \mathbf{t})$, and encoder $q_{\boldsymbol{r}, \boldsymbol{s}}(\mathbf{z}|\mathbf{x}, \mathbf{y}, \mathbf{t})$ are factorized Gaussians, i.e., a product of 1-dimensional Gaussian distributions. This is not restrictive if the mean and variance are given by arbitrary nonlinear functions.

$$\mathbf{y}|\mathbf{z}, \mathbf{t} \sim \prod_{j=1}^{d} \mathcal{N}(y_j; f_j, g_j), \quad \mathbf{z}|\mathbf{x}, \mathbf{t} \sim \prod_{i=1}^{n} \mathcal{N}(z_i; h_i, k_i), \quad \mathbf{z}|\mathbf{x}, \mathbf{y}, \mathbf{t} \sim \prod_{i=1}^{n} \mathcal{N}(z_i; r_i, s_i).$$
$$(7)$$

$\boldsymbol{\theta} = (\boldsymbol{f}, \boldsymbol{g}, \boldsymbol{h}, \boldsymbol{k})$ and $\boldsymbol{\phi} = (\boldsymbol{r}, \boldsymbol{s})$ are functional parameters given by NNs which take the respective conditional variables as inputs (e.g. $\boldsymbol{h} := (h_i(\boldsymbol{x}, t))^T$).

## 5 IDENTIFYING REPRESENTATION AND TREATMENT EFFECTS

In the following, we will show that CFVAE can identify the latent variable up to an affine transformation (Sec. 5.1), it can learn a balancing score as a causal representation, and its decoder is a valid estimator for potential outcomes (Sec. 5.2).

### 5.1 IDENTIFIABILITY OF REPRESENTATION

In this subsection, we show that CFVAE can identify latent variable $\mathbf{z}$ up to an element-wise affine transformation when the noise on the outcome is small. Based on this result, we can gain insight on how to make CFVAE learn a balancing score.

Our starting point is the following theorem showing the identifiability of our learning model, adapted from Theorem 1 in Khemakhem et al. (2020), by adding conditioning on $\mathbf{t}$.

**Theorem 2.** *Given the family $p_{\boldsymbol{\theta}}(\mathbf{y}, \mathbf{z}|\mathbf{x}, \mathbf{t})$ specified by (5) and (7)[2], for $t = 0, 1$, assume*

*1) $\boldsymbol{f}_t(\boldsymbol{z}) := (f_i(\boldsymbol{z}, t))^T$ is injective;*

*2) $\boldsymbol{g}_t(\boldsymbol{z}) = \boldsymbol{\sigma}_{\mathbf{y}, t}$ is constant (i.e. $g_i(\boldsymbol{z}, t) = \sigma_{y_i, t}$);*

*3) $\boldsymbol{\lambda}_t(\mathbf{x}) := (\boldsymbol{h}(\mathbf{x}, t), \boldsymbol{k}(\mathbf{x}, t))^T$, which is seen as a random variable, is not degenerate.*

---

[2]We specified factorized Gaussians in (7) and they show good performance in our experiments. But Corollary 1 and Theorem 3 can be extended to more general exponential families, see Khemakhem et al. (2020).

*Then, given* $t = t$, *the family is identifiable up to an equivalence class. That is, for* $t = 0, 1$, *if* $p_{\boldsymbol{\theta}_t}(\mathbf{y}|\mathbf{x}, t = t) = p_{\boldsymbol{\theta}'_t}(\mathbf{y}|\mathbf{x}, t = t)^3$, *we have the relation between parameters*

$$\boldsymbol{f}_t^{-1}(\mathbf{y}_t) = \boldsymbol{A}_t \boldsymbol{f}'^{-1}_t(\mathbf{y}_t) + \boldsymbol{b}_t \coloneqq \mathcal{A}_t(\boldsymbol{f}'^{-1}_t(\mathbf{y}_t)) \tag{8}$$

*where* $p(\mathbf{y}_t) \coloneqq p(\mathbf{y}|t)$, $\boldsymbol{A}_t$ *is an invertible* $n$-*square matrix and* $\boldsymbol{b}_t$ *is a* $n$-*vector.*

Similarly to Sorrenson et al. (2019), we can further show that $\boldsymbol{A}_t = \mathrm{diag}(\boldsymbol{a}_t)$ is a diagonal matrix. By a slight abuse of symbol, we will overload $|$ to make a shorthand for equations like (8), e.g., (8) can be written as $\boldsymbol{f}^{-1}(\mathbf{y}) = \mathcal{A}(\boldsymbol{f}'^{-1}(\mathbf{y}))|t$. Note that, by definition of inverse, we also have $\boldsymbol{f}' = \boldsymbol{f} \circ \mathcal{A}|t$.

The importance of model identifiability can be seen more clearly in the limit of small noise on $\mathbf{y}$. Corollary 1 can be easily understood by noting that after learning with small noise on $\mathbf{y}$, the encoder and decoder both degenerate to deterministic functions: in (7) $\boldsymbol{g}' = \boldsymbol{s}' = \mathbf{0}$, and $\forall \boldsymbol{x}, \boldsymbol{z}'_t = \boldsymbol{r}'_t(\boldsymbol{x}, \boldsymbol{y}) = \boldsymbol{f}'^{-1}_t(\boldsymbol{y})$. Note that, we only assume the VAE learns *observational* distributions $p_{\boldsymbol{\theta}'_t}(\mathbf{y}|\mathbf{x}, t = t)$ the same as the truth, but this leaves room for *latent* distributions different to the truth.

**Corollary 1** (Identifiability of representation). *For* $t = 0, 1$, *assume 1)* $\boldsymbol{\sigma}_{\mathbf{y},t} \to \mathbf{0}$ *and 2) CFVAE can learn a distribution* $p_{\boldsymbol{\theta}'_t} = p_{\boldsymbol{\theta}_t}$, *then the latent variable* $\mathbf{z}$ *and the mean parameter* $\boldsymbol{f}_t$ *of* $\mathbf{y}$ *can be identified up to an element-wise affine transformation:* $\mathbf{z} = \mathcal{A}(\mathbf{z}')|t$ *and* $\boldsymbol{f}' = \boldsymbol{f} \circ \mathcal{A}|t$.

This is a strong result for learning interpretable representation, but it is not enough for causal inference. To see this in a principled way, we recall the concept of balancing score. The recovered latent $\mathbf{z}'$ in Corollary 1 is not a balancing score, due to the different $\mathcal{A}_t$ for $t = 0, 1$. If $\mathbf{z}'$ *were* a balancing score, we would have $t \perp\!\!\!\perp \mathbf{z}|\mathbf{z}'$. However, given $\mathbf{z}' = \mathbf{z}'$, $\mathbf{z} = \mathrm{diag}(\boldsymbol{a}_t)\mathbf{z}' + \boldsymbol{b}_t$ is a deterministic function of $t$, contradicting with $t \perp\!\!\!\perp \mathbf{z}|\mathbf{z}'$. (A more concrete analysis can be found in Appendix.) This example also suggests that we will have a balancing score if we can get rid of the dependence on $t = t$. The next subsection discusses some assumptions on $\mathbf{x}$ to remove the "$|t$" in Corollary 1.

## 5.2 IDENTIFICATION OF TREATMENT EFFECTS

The following definition will be used in Theorem 3. The importance of this definition is immediate from Theorem 1, that is, if a balancing *covariate* is also a function of $\mathbf{z}$, then it is a balancing *score*.

**Definition 1** (Balancing covariate). *Random variable* $\mathbf{x}$ *is a* balancing covariate *of random variable* $\mathbf{z}$ *if* $t \perp\!\!\!\perp \mathbf{z}|\mathbf{x}$. *We also simply say* $\mathbf{x}$ *is* balancing *(or* non-*balancing if it does not satisfy this definition).*

Given that a balancing score of the true confounder is sufficient for strong ignorability, a natural and interesting question is that, does a balancing covariate of the true confounder also satisfies strong ignorability? The answer is *no*. To see why, and also to better understand the significance of Theorem 3, we give Proposition 1 indicating that a balancing covariate of the true confounder might *not* even satisfy *ignorability*. We also refer readers to Appendix 8.5 where we examine two important special cases of balancing covariate, one of those is *noiseless proxy*, which might *not* satisfy *positivity*.

**Proposition 1.** *Let* $\mathbf{x}$ *be a balancing covariate of* $\mathbf{z}$. *If* $\mathbf{z}$ *satisfies ignorability and* $y(0), y(1) \perp\!\!\!\perp \mathbf{x}|\mathbf{z}, t$, *then* $\mathbf{x}$ *satisfies ignorability.*

Given this proposition, we know our assumptions are weaker than ignorability, and our method can work under unobserved confounding ($\mathbf{x}$ might not satisfy ignorability). Note the independence $y(0), y(1) \perp\!\!\!\perp \mathbf{x}|\mathbf{z}, t$ in this proposition *implies* $y \perp\!\!\!\perp \mathbf{x}|\mathbf{z}, t$ assumed by CFVAE. From the decomposition rule of conditional independence, we have $y(0), y(1) \perp\!\!\!\perp \mathbf{x}|\mathbf{z}, t \implies \forall t (y(t) \perp\!\!\!\perp \mathbf{x}|\mathbf{z}, t) \implies \forall t, \bar{t} (y(t) \perp\!\!\!\perp \mathbf{x}|\mathbf{z}, t = \bar{t})$, 4 independence in total. Using only two of them, and with the consistency of counterfactual, we have $\forall t (y(t) \perp\!\!\!\perp \mathbf{x}|\mathbf{z}, t = t) \implies \forall t (y \perp\!\!\!\perp \mathbf{x}|\mathbf{z}, t = t) \implies y \perp\!\!\!\perp \mathbf{x}|\mathbf{z}, t$. In general, the other two independence $y(0) \perp\!\!\!\perp \mathbf{x}|\mathbf{z}, t = 1$ and $y(1) \perp\!\!\!\perp \mathbf{x}|\mathbf{z}, t = 0$ do not hold, and thus $y \perp\!\!\!\perp \mathbf{x}|\mathbf{z}, t \centernot\implies y(0), y(1) \perp\!\!\!\perp \mathbf{x}|\mathbf{z}, t$.

As shown in the following theorem, for any balancing covariates, CFVAE can identify treatment effects, by learning a balancing score of the true confounder as latent representation. Note that

---

$^3 \boldsymbol{\theta}'_t = (\boldsymbol{f}'_t, \boldsymbol{h}'_t, \boldsymbol{k}'_t)$ is another set of parameters giving the same distribution, which is learned by VAE. In this paper, symbol "$'$" (prime) always indicates parameters (variables, etc.) learned/recovered by VAE.

assumption 3) is nontrivial, it requires that the true data generating distribution is in the learning model. And thus identification of treatment effects follows from the identifiability of our model.

**Theorem 3** (Identification with balancing covariate). *Assume*

*1) the same as Theorem 2 and Corollary 1;*

*2) $h, k$ in (7) depend on $\mathbf{x}$ but not $\mathbf{t}$ (i.e. $h_i(\boldsymbol{x}, t) = h_i(\boldsymbol{x})$ and same for $k$);*

*3) The data generating process can be parametrized by family $p_{\boldsymbol{\theta}}(\mathbf{y}, \mathbf{z}|\mathbf{x}, \mathbf{t})$ specified above;*

*4) $\mathbf{z}$ satisfies strong ignorability and $\mathbf{x}$ is a balancing covariate of $\mathbf{z}$.*

*Then, for both $t = 0, 1$, we have $\mathbf{z} = \mathrm{diag}(\boldsymbol{a})\mathbf{z}' + \boldsymbol{b} := \mathcal{A}(\mathbf{z}')$, and $\mathbf{z}'$ satisfies strong ignorability. We identify the potential outcomes by $\mu_{\bar{t}}(\boldsymbol{x}) = \mathbb{E}(\mathbb{E}(\mathbf{y}|\mathbf{z}', \mathbf{x} = \boldsymbol{x}, \mathbf{t} = \bar{t})) = \mathbb{E}(\boldsymbol{f}'_{\bar{t}}(\boldsymbol{r}'_t(\boldsymbol{x}, \mathbf{y}_t))|\mathbf{x} = \boldsymbol{x}).$*

The result may be more easily understood as following. Now with the same $\mathcal{A}$ for both treatment groups $t$, given observation $y_t$, the counterfactual prediction given by CFVAE is the same as truth: $y'_{1-t} = \boldsymbol{f}'_{1-t}(\boldsymbol{z}'_t) = \boldsymbol{f}_{1-t} \circ \mathcal{A}(\mathcal{A}^{-1}(\boldsymbol{z}_t)) = \boldsymbol{f}_{1-t}(\boldsymbol{z}_t) = y(1-t)$ (Also compare this to Appendix 8.4). We can identify potential outcomes, using $\boldsymbol{r}'_t(\boldsymbol{x}, y_t) = \boldsymbol{z}'_t$ in the last equality, by

$$\mu_{\bar{t}}(\boldsymbol{x}) = \mathbb{E}(\mathbf{y}(\bar{t})|\mathbf{x} = \boldsymbol{x}) = \mathbb{E}(\mathbf{y}'_{\bar{t}}|\mathbf{x} = \boldsymbol{x}) = \mathbb{E}(\boldsymbol{f}'_{\bar{t}}(\boldsymbol{r}'_t(\boldsymbol{x}, \mathbf{y}_t))|\mathbf{x} = \boldsymbol{x}) \quad (9)$$

Note that counterfactual assignment $\mathbf{t} = \bar{t}$ may or may not be the same as factual $t$. The algorithm for estimation CATE and ATE is as following. After training CFVAE, we feed data $\mathcal{D} = \{(\boldsymbol{x}, y_t) := (\boldsymbol{x}, y, t)\}$ into the encoder, and draw sample from it: $q(\mathbf{z}'|\mathbf{x} = \boldsymbol{x}, \mathbf{y} = y, \mathbf{t} = t) = \delta(\mathbf{z}' - \boldsymbol{r}'_t(\boldsymbol{x}, y_t))$ ($\delta$ denotes delta function). Then, setting $\mathbf{t} = \bar{t} \in \{0, 1\}$ in the decoder, feed posterior sample $\{\boldsymbol{z}'_t = \boldsymbol{r}'_t(\boldsymbol{x}, y_t)\}$, we get counterfactual prediction $p(\mathbf{y}'|\mathbf{z}' = \boldsymbol{z}'_t, \mathbf{t} = \bar{t}) = \delta(\mathbf{y}'_{\bar{t}} - \boldsymbol{f}'_{\bar{t}}(\boldsymbol{z}'_t))$. Finally, we estimate ATE by taking average $\mathbb{E}_{\mathcal{D}}(y'_1 - y'_0)$, and CATE by $\mathbb{E}_{\{\mathcal{D}|\mathbf{x}=\boldsymbol{x}\}}(y'_1 - y'_0)$, adding conditioning on $\boldsymbol{x}$.

A caveat is that (9) requires *post-treatment* observation $y_t$. Often, it is desirable that we can also have *pre-treatment* prediction for a new subject, with only the observation of its covariate $\mathbf{x} = \boldsymbol{x}$. To this end, we use conditional prior $p(\mathbf{z}'|\mathbf{x})$ as a pre-treatment predictor for $\mathbf{z}'$: input $\boldsymbol{x}$ and draw sample from $p(\mathbf{z}'|\mathbf{x} = \boldsymbol{x})$ instead of $q$, and all the others remain the same. Since the ELBO has a KL term between $p(\mathbf{z}'|\mathbf{x})$ and $q$, the two distributions should not be very different, and we will also have sensible pre-treatment estimation of treatment effects.

Although our method works under unobserved confounding, it still formally requires small outcome noise and balancing covariate. However, experiments show our method can work very well with large outcome noise, and the covariates can be non-balancing and also directly affect the outcome, including general proxies, IVs, and even networked data.

## 6 EXPERIMENTS

As in previous works (Shalit et al., 2017; Louizos et al., 2017), we report the absolute error of ATE $\epsilon_{ATE} := |\mathbb{E}_{\mathcal{D}}(y(1) - y(0)) - \mathbb{E}_{\mathcal{D}}(y'_1 - y'_0)|$, and the square root of empirical PEHE (Hill, 2011) $\epsilon_{PEHE} := \mathbb{E}_{\mathcal{D}}((y(1) - y(0)) - (y'_1 - y'_0))^2$ for individual-level treatment effects.

Unless otherwise indicated, for each function $\boldsymbol{f}, \boldsymbol{g}, \boldsymbol{h}, \boldsymbol{k}, \boldsymbol{r}, \boldsymbol{s}$ in (7), we use a multilayer perceptron (MLP) that has 3*200 hidden units with ReLU activation, and $\boldsymbol{h}, \boldsymbol{k}$ depend only on $\mathbf{x}$. The Adam optimizer with initial learning rate $10^{-4}$ and batch size 100 is employed. More details on hyperparameters and experimental settings are given in each experiment, and are explained in Appendix.

All experiments use early-stopping of training by evaluating the ELBO on a validation set. We evaluate the post-treatment performance on training and validation set jointly (This is non-trivial. Recall the fundamental problem of causal inference). The treatment and (factual) outcome should not be observed for pre-treatment predictions, so we report them on a testing set.

### 6.1 SYNTHETIC DATASET

We generate data following (10) with z, y 1-dimensional and x 3-dimensional. $\mu_i$ and $\sigma_i$ are randomly generated in range $(-0.2, 0.2)$ and $(0, 0.2)$, respectively. The functions $h, k, l$ are linear with random coefficients. The outcome model is built for the two treatments separately, i.e.

$f(\mathbf{z}, t) := f_t(\mathbf{z}), t = 0, 1$. We generate two kinds of outcome models, depending on the type of $f_t$: linear and nonlinear outcome models use random linear functions and NNs with random weights, respectively.

$$\mathbf{x} \sim \prod_{i=1}^{3} \mathcal{N}(\mu_i, \sigma_i); \quad \mathbf{z}|\mathbf{x} \sim \mathcal{N}(h(\mathbf{x}), \beta k(\mathbf{x}));$$
$$\mathbf{t}|\mathbf{x}, \mathbf{z} \sim \text{Bern}(\text{Logistic}(l(\mathbf{x}, \mathbf{z}))); \quad \mathbf{y}|\mathbf{z}, \mathbf{t} \sim \mathcal{N}(C_t^{-1} f(\mathbf{z}, \mathbf{t}), \alpha). \tag{10}$$

We adjust the outcome and proxy noise level by $\alpha, \beta$ respectively. The output of $f_t$ is normalized by $C_t := \text{Var}_{\{\mathcal{D}|t=t\}}(f_t(\mathbf{z}))$. This means we need to use $0 \leq \alpha < 1$ to have a reasonable level of noise on $\mathbf{y}$ (the scales of mean and variance are comparable). Similar reasoning applies to $\mathbf{z}|\mathbf{x}$; outputs of $h, k$ have approximately the same range of values since the functions' coefficients are generated by the same weight initializer.

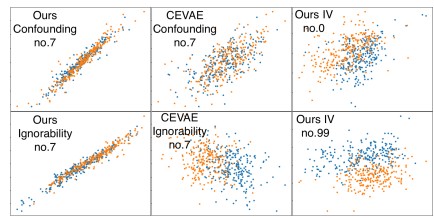

We experiment on three different causal settings (indicated in *Italic*). To introduce $\mathbf{x}$ as *IV*, we generate another 1-dimensional random source $\mathbf{w}$ in the same way as $\mathbf{x}$, and use $\mathbf{w}$ instead of $\mathbf{x}$ to generate $\mathbf{z}|\mathbf{w} \sim \mathcal{N}(h(\mathbf{w}), \beta k(\mathbf{w}))$. Besides taking inputs $\mathbf{x}, \mathbf{z}$ in $l$, we consider two special cases: $l := l(\mathbf{x})$ ($\mathbf{x}$ fully satisfies *ignor-ability*) and $l := l(\mathbf{z})$ (unobserved confounder $\mathbf{z}$ and *non-balancing proxy* $\mathbf{x}$ of $\mathbf{z}$). Except indicated above, other aspects of the models are specified by (10). See Appendix for graphical models of these three cases.

Figure 2: Plots of recovered (x) - true (y) latent on the nonlinear outcome. Blue: $t = 0$, Orange: $t = 1$. $\alpha, \beta = 0.4$. "no." indicates index among the 100 random models.

In each causal setting, and with the same kind of outcome models and noise levels $(\alpha, \beta)$, we evaluate CFVAE and CEVAE on 100 random data generating models, with different sets of functions $f, h, k, l$ in (10). For each model, we sample 1500 data points, and split them into 3 equal sets for training, validation, and testing. Both the methods use 1-dimensional latent variable in VAE. For fair comparison, all the hyper-parameters, including type and size of NNs, learning rate, and batch size, are the same for both the methods.

Figure 3 shows our method significantly outperforms CE-VAE on all cases. Each method works the best under ignorability, as expected. The performances of our method on IV and proxy settings match that of CEVAE under ignorability, showing the effective deconfounding. Figure 2 shows our method learns highly interpretable representation as an approximate affine transformation of the true latent value. To our surprise, CEVAE is also possible to achieve this when both noises are small, though the quality of recovery is lower than CFVAE. The relationship to the true latent is significantly obscured under IVs, because the true latent is correlated to IVs only given t, while we model it by $p(\mathbf{z}'|\mathbf{x})$ as required by Theorem 3.

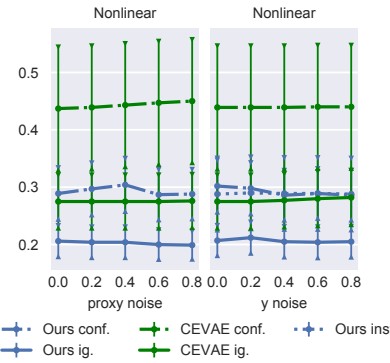

We can see our method and also CEVAE are very robust w.r.t. both outcome and proxy noise. This may due to the good probabilistic modeling of the noise by VAE. Still, we can see in Appendix that the noise level affects how well we recover the latent variable.

Figure 3: Pre-treatment $\sqrt{\epsilon_{PEHE}}$ on non-linear synthetic dataset. Error bar on 100 random models. We adjust one of the noise levels $\alpha, \beta$ in each panel, with another fixed to 0.2. See Appendix for results on linear outcome. Results for ATE and post-treatment are similar.

## 6.2 IHDP BENCHMARK DATASET

The IHDP dataset (Hill, 2011) is widely used to evaluate machine learning based causal inference methods, e.g. Shalit et al. (2017); Shi et al. (2019). Here, *ignorability* holds given the covariates. See Appendix for detailed descriptions. Note, however, that this dataset violates our assumption $\mathbf{y} \perp\!\!\!\perp \mathbf{x}|\mathbf{z}, \mathbf{t}$, since the covariates $\mathbf{x}$ directly affect the outcome. To overcome this, we add two components introduced by Shalit et al. (2017) into our method. First, we build two outcome functions $\boldsymbol{f}_t(\mathbf{z}), t = 0, 1$ in our learning model (7), using two separate NNs.

Second, we add to our ELBO (6) a regularization term, which is the Wasserstein distance (Cuturi, 2013) between the learned $p(\mathbf{z}'|\mathbf{x}, \mathrm{t} = 0)$ and $p(\mathbf{z}'|\mathbf{x}, \mathrm{t} = 1)$. We find higher than 1-dimensional latent variable in CFVAE gives better results, very possibly due to the mismatched latent distribution: the confounder `race` is discrete but we use Gaussian latent variable. We report results with 10-dimensional latent variable.

As shown in Table 1, the proposed CFVAE matches the state-of-the-art methods under model misspecification. This robustness of VAE was also observed by Louizos et al. (2017), where they used 5-dimensional Gaussian latent variable to model a binary ground truth. And notably, without the two additional modifications, our method has the best ATE estimation and is overall the best among generative models (better than CEVAE and GANITE by a large margin).

Table 1: Errors on IHDP. "A/B" means pre-treatment/post-treatment prediction. The mean and std are calculated over 1000 random draws of the data generating model. *Results with the two modifications. The results without the modifications are $\epsilon_{ATE} = .21_{\pm.01}/.17_{\pm.01}$ and $\sqrt{\epsilon_{PEHE}} = 1.0_{\pm.05}/.97_{\pm.04}$. **Bold** indicates method(s) that are *significantly* better than all the others. The results of the other methods are taken from Shalit et al. (2017), except GANITE (Yoon et al., 2018) and CEVAE (Louizos et al., 2017).

| Method | TMLE | BNN | CFR | CF | CEVAE | GANITE | Ours* |
|---|---|---|---|---|---|---|---|
| $\epsilon_{ATE}$ | NA/.30$_{\pm.01}$ | .42$_{\pm.03}$/.37$_{\pm.03}$ | .27$_{\pm.01}$/.25$_{\pm.01}$ | .40$_{\pm.03}$/**.18**$_{\pm.01}$ | .46$_{\pm.02}$/.34$_{\pm.01}$ | .49$_{\pm.05}$/.43$_{\pm.05}$ | .31$_{\pm.01}$/.30$_{\pm.01}$ |
| $\sqrt{\epsilon_{PEHE}}$ | NA/5.0$_{\pm.2}$ | 2.1$_{\pm.1}$/2.2$_{\pm.1}$ | **.76**$_{\pm.02}$/**.71**$_{\pm.02}$ | 3.8$_{\pm.2}$/3.8$_{\pm.2}$ | 2.6$_{\pm.1}$/2.7$_{\pm.1}$ | 2.4$_{\pm.4}$/1.9$_{\pm.4}$ | **.77**$_{\pm.02}$/**.69**$_{\pm.02}$ |

## 6.3 POKEC SOCIAL NETWORK DATASET

Pokec (Leskovec & Krevl, 2014) is a real world social network dataset. We experiment on a semi-synthetic dataset based on Pokec, which was introduced in Veitch et al. (2019), and use exactly the same pre-processing and generating procedure. The pre-processed network has about 79,000 vertexes (users) connected by $1.3 \times 10^6$ undirected edges. The subset of users used here are restricted to three living districts that are within the same region. The network structure is expressed by binary adjacency matrix $\boldsymbol{G}$. Following Veitch et al. (2019), we split the users into 10 folds, test on each fold and report the mean and std of pre-treatment ATE predictions. We further separate the rest of users (in the other 9 folds) by $6 : 3$, for training and validation. Table 2 shows the results. Our method is the best compared with the methods specialized for networked data. We report pre-treatment PEHE of our method in the Appendix, while Veitch et al. (2019) does not give individual-level prediction.

Table 2: Pre-treatment ATE on Pokec. Ground truth is 1. "Unadjusted" estimates ATE by $\mathbb{E}_{\mathcal{D}}(y_1) - \mathbb{E}_{\mathcal{D}}(y_0)$. "Parametric" is a stochastic block model for networked data (Gopalan & Blei, 2013). "Embed-" denotes the best alternatives given by Veitch et al. (2019). **Bold** indicates method(s) that are *significantly* better than all the others. 20-dimensional latent variable in CFVAE works better, and its result is reported. The results of the other methods are taken from Veitch et al. (2019).

| | Unadjusted | Parametric | Embed-Reg. | Embed-IPW | Ours |
|---|---|---|---|---|---|
| Age | $4.34 \pm 0.05$ | $4.06 \pm 0.01$ | $2.77 \pm 0.35$ | $3.12 \pm 0.06$ | **2.08** $\pm 0.32$ |
| District | $4.51 \pm 0.05$ | $3.22 \pm 0.01$ | **1.75** $\pm 0.20$ | **1.66** $\pm 0.07$ | **1.68** $\pm 0.10$ |
| Join Date | $4.03 \pm 0.06$ | $3.73 \pm 0.01$ | $2.41 \pm 0.45$ | $3.10 \pm 0.07$ | **1.70** $\pm 0.13$ |

Each user has 12 attributes, among which `district`, `age`, or `join date` is used as a confounder z to build 3 different datasets, with remaining 11 attributes used as covariate $\mathbf{x}$. Treatment t and outcome y are synthesised as following:

$$\mathrm{t} \sim \mathrm{Bern}(g(\mathrm{z})); \quad \mathrm{y} = \mathrm{t} + 10(g(\mathrm{z}) - 0.5) + \epsilon, \epsilon \sim \mathcal{N}(0, 1) \tag{11}$$

Note that `district` is of 3 categories; `age` and `join date` are also discretized into three bins. $g(\mathrm{z})$ maps these three categories and values to $\{0.15, 0.5, 0.85\}$.

Some assumptions to justify our method may not hold in this dataset. The important challenges are 1) $\mathbf{x}$ obviously does not satisfy ignorability, and 2) large outcome noise exists. On the other hand, given the huge network structure, most users can practically be identified by their attributes and neighborhood structure, which means z can be roughly seen as a deterministic function of $\boldsymbol{G}, \mathbf{x}$. Then, $\boldsymbol{G}, \mathbf{x}$ can be, as defined by us, *noiseless proxies* of z (see Appendix 8.5). CFVAE is then expected to control for the confounding to a large extent and able to learn a balancing score based on

Theorem 3, if we can exploit the network structure effectively. This idea is comparable to Assumptions 2 and 4 in Veitch et al. (2019), which postulate directly that a balancing score can be learned in the limit of infinite large network.

To extract information from the network structure, we use Graph Convolutional Network (GCN) (Kipf & Welling, 2017) in conditional prior and encoder of CFVAE. A difficulty is that, the network $G$ and covariates $X$ of *all* users are always needed by GCN, regardless of whether it is in training, validation, or testing phase. However, the separation can still make sense if we take care that the treatment and outcome are used only in the respective phase, e.g., $(y_m, t_m)$ of a testing user $m$ is only used in testing. See Appendix for details.

## 7    DISCUSSION

In this work, we proposed a new VAE architecture for estimating causal effects under unobserved confounding, with theoretical analysis and state-of-the-art performance. To the best of our knowledge, this is the *first* generative learning method that *provably* identifies treatment effects, without directly assuming that the true latent variable can be recovered. It is achieved by, on the one hand, noticing we only need the part of latent information that is correlated to treatment assignment, and, on the other hand, exploiting the recent advances that the latent variable can be recovered up to trivial transformations in a broad class of generative models.

Despite the formal requirement, the experiments show our method is robust to large outcome noise. Theoretical analysis of this phenomenon is an interesting direction for future work. A related theoretical issue is that, while Khemakhem et al. (2020) assumes fixed distribution of noise on $\mathbf{y}$, we observed that, in most cases, allowing the noise distribution to depend on $\mathbf{z}, \mathbf{t}$ improves performance. Extending identifiability to conditional noise models is also an interesting direction.

When the latent model is misspecified (Sec. 6.2 and 6.3), our method still matches the state-of-the-art, though we cannot see apparent relationship between recovered latent variable and the true one. It would be nice to see the learned representation indeed preserves causal properties under model misspecification, for example, by some causally-specialized metrics, e.g. Suter et al. (2019). Given the fact that all nonlinear ICA based identifiability requires an injective mapping between the latent and observed variables, theoretical extensions to discrete latent variable would be challenging.

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

# 8 APPENDIX

## 8.1 PROOFS

*Proof of Corollary 1.* We need the *consistency*[4] of our VAE to learn a observational distribution equaling to the true one in the limit of infinite data, so that the learned parameters $\boldsymbol{\theta}_t'$ is in the equivalence class of $\boldsymbol{\theta}_t$ defined by (8). This can be proved (Khemakhem et al., 2020, Theorem 4) by assuming: 1) our VAE is flexible enough to ensure the ELBO is tight (equals to the log likelihood of our model) for some parameters; 2) the optimization algorithm can achieve the *global* maximum of ELBO (again equals to the log likelihood).

In this proof, all equations and variables should condition on $t$, and we omit the conditioning in notation for convenience. In the limit of $\boldsymbol{\sigma_y} \to \mathbf{0}$, the decoder degenerates to a delta function: $p(\mathbf{y}|\mathbf{z}) = \delta(\mathbf{y} - \boldsymbol{f}(\mathbf{z}))$, we have $\mathbf{y} = \boldsymbol{f}(\mathbf{z})$ and $\mathbf{y}' = \boldsymbol{f}'(\mathbf{z}')$. From the consistency of VAE, $\mathbf{y}$ should have the same support as $\mathbf{y}'$. For all $\boldsymbol{y}$ in the support, there exist a *unique $z$* and a *unique $z'$* satisfy $\boldsymbol{y} = \boldsymbol{f}(\boldsymbol{z}) = \boldsymbol{f}'(\boldsymbol{z}')$ (use injectivity). Substitute $\boldsymbol{y} = \boldsymbol{f}(\boldsymbol{z})$ into the l.h.s of (8), and $\boldsymbol{y} = \boldsymbol{f}'(\boldsymbol{z}')$ into the r.h.s, we have $\boldsymbol{z} = \mathrm{diag}(\boldsymbol{a})\boldsymbol{z}' + \boldsymbol{b}$. The relation is one-to-one for all $\boldsymbol{z}$, so we get $\mathbf{z} = \mathcal{A}(\mathbf{z}')$. Similar result for $\boldsymbol{f}$ follows. $\qquad\square$

**Proposition 2** (Properties of conditional independence). *For random variables* $\mathbf{w}, \mathbf{x}, \mathbf{y}, \mathbf{z}$. *We have (Pearl, 2009, 1.1.55):*

$$\mathbf{x} \perp\!\!\!\perp \mathbf{y}|\mathbf{z} \wedge \mathbf{x} \perp\!\!\!\perp \mathbf{w}|\mathbf{y}, \mathbf{z} \implies \mathbf{x} \perp\!\!\!\perp \mathbf{w}, \mathbf{y}|\mathbf{z} \text{ (Contraction)}.$$
$$\mathbf{x} \perp\!\!\!\perp \mathbf{w}, \mathbf{y}|\mathbf{z} \implies \mathbf{x} \perp\!\!\!\perp \mathbf{y}|\mathbf{w}, \mathbf{z} \text{ (Weak union)}.$$
$$\mathbf{x} \perp\!\!\!\perp \mathbf{w}, \mathbf{y}|\mathbf{z} \implies \mathbf{x} \perp\!\!\!\perp \mathbf{y}|\mathbf{z} \text{ (Decomposition)}.$$

*Proof of Proposition 1.* This proof will use the above three properties of conditional independence. We first write our assumptions in conditional independence, as *A1.* $\mathsf{t} \perp\!\!\!\perp \mathbf{z}|\mathbf{x}$ (balancing covariate), *A2.* $\mathbf{w} \perp\!\!\!\perp \mathsf{t}|\mathbf{z}$ (ignorability given $\mathbf{z}$), and *A3.* $\mathbf{w} \perp\!\!\!\perp \mathbf{x}|\mathbf{z}, \mathsf{t}$, where $\mathbf{w} := (\mathsf{y}(0), \mathsf{y}(1))$.

Now, from *A2* and *A3*, using contraction, we have $\mathbf{w} \perp\!\!\!\perp \mathbf{x}, \mathsf{t}|\mathbf{z}$, then using weak union, we have $\mathbf{w} \perp\!\!\!\perp \mathsf{t}|\mathbf{x}, \mathbf{z}$. From this last independence and *A1*, using contraction, we have $\mathsf{t} \perp\!\!\!\perp \mathbf{z}, \mathbf{w}|\mathbf{x}$. Then $\mathsf{t} \perp\!\!\!\perp \mathbf{w}|\mathbf{x}$ follows by decomposition. $\qquad\square$

*Proof of Theorem 2.* From the proof of Theorem 1 in Khemakhem et al. (2020), we know $\boldsymbol{A}_t, \boldsymbol{b}_t$ depend on $t$ only through $\boldsymbol{h}, \boldsymbol{k}$. But we assume $\boldsymbol{h}, \boldsymbol{k}$ do not depend on t. So we have $\mathbf{z} = \mathcal{A}(\mathbf{z}')$ for both $t = 0, 1$. From Theorem 1, $\mathbf{z}'$ is a balancing score of $\mathbf{z}$, and satisfies strong ignorability. Here, we proceed a bit different from (2), we have

$$\mu_{\bar{t}}(\boldsymbol{x}) = \mathbb{E}(\mathbb{E}(\mathsf{y}(\bar{t})|\mathbf{z}', \mathbf{x} = \boldsymbol{x})) = \int \mathbb{E}(\mathsf{y}(\bar{t})|\mathbf{z}' = \boldsymbol{z}', \mathbf{x} = \boldsymbol{x})p(\boldsymbol{z}'|\mathbf{x} = \boldsymbol{x})d\boldsymbol{z}'$$

$$= \int \mathbb{E}(\mathsf{y}|\mathbf{z}' = \boldsymbol{z}', \mathbf{x} = \boldsymbol{x}, \mathsf{t} = \bar{t})p(\boldsymbol{z}'|\mathbf{x} = \boldsymbol{x})d\boldsymbol{z}' = \int (\int p(y|\boldsymbol{z}', \boldsymbol{x}, \bar{t})y\,dy)p(\boldsymbol{z}'|\boldsymbol{x})d\boldsymbol{z}' \tag{12}$$

Compare the rightmost side to (2), note that, there is no conditioning on t in $p(\boldsymbol{z}'|\boldsymbol{x})$, because we use the strong ignorability given $\mathbf{z}'$ (and consistency of counterfactuals) in the third equality, *after* expanding the outer expectation.

From the consistency of VAE, $p(\mathsf{y}|\boldsymbol{z}', \boldsymbol{x}, \bar{t}) = \delta(\mathsf{y} - \boldsymbol{f}_{\bar{t}}'(\boldsymbol{z}'))$, and $q(\boldsymbol{z}'|\boldsymbol{x}, y, t) = \delta(\boldsymbol{z}' - \boldsymbol{r}_t'(\boldsymbol{x}, y_t)) = p(\boldsymbol{z}'|\boldsymbol{x}, y, t) = \delta(\boldsymbol{z}' - \boldsymbol{f}_t'^{-1}(y_t))$ where $(\boldsymbol{x}, y_t) := (\boldsymbol{x}, y, t)$ is a data point. And $p(\boldsymbol{z}'|\boldsymbol{x}) = \sum_t \int_y p(\boldsymbol{z}'|\boldsymbol{x}, y, t)p(y, t|\boldsymbol{x})dy = \sum_t \int_y \delta(\boldsymbol{z}' - \boldsymbol{f}_t'^{-1}(y_t))p(y, t|\boldsymbol{x})dy$. We have

$$\mu_{\bar{t}}(\boldsymbol{x}) = \int_{\boldsymbol{z}'} (\boldsymbol{f}_{\bar{t}}'(\boldsymbol{z}') \sum_t \int_y \delta(\boldsymbol{z}' - \boldsymbol{f}_t'^{-1}(y_t))p(y, t|\boldsymbol{x})dy)d\boldsymbol{z}'$$

$$= \sum_t \int_y \boldsymbol{f}_{\bar{t}}'(\boldsymbol{f}_t'^{-1}(y_t))p(y, t|\boldsymbol{x})dy = \mathbb{E}(\boldsymbol{f}_{\bar{t}}'(\boldsymbol{r}_t'(\boldsymbol{x}, y_t))|\mathbf{x} = \boldsymbol{x}) \tag{13}$$

---

[4]This is the statistical consistency of an estimator. Do not confuse with the consistency of counterfactuals.

We should note that $p(\mathbf{z}'|\boldsymbol{x}, y, t) := p_{\boldsymbol{\theta}'_t}(\mathbf{z}'|\mathbf{x} = \boldsymbol{x}, \mathbf{y} = y, \mathbf{t} = t) = p_{\boldsymbol{\theta}'_t}(\mathbf{y} = y, \mathbf{z}'|\mathbf{x} = \boldsymbol{x}, \mathbf{t} = t)/\int p_{\boldsymbol{\theta}'_t}(\mathbf{y} = y, \mathbf{z}'|\mathbf{x} = \boldsymbol{x}, \mathbf{t} = t)d\mathbf{z}'$ might *not* be equal to the truth $p(\mathbf{z}|\boldsymbol{x}, y, t) := p_{\boldsymbol{\theta}_t}(\mathbf{z}|\mathbf{x} = \boldsymbol{x}, \mathbf{y} = y, \mathbf{t} = t)$ (in particular it is possible that $\boldsymbol{f}'_t \neq \boldsymbol{f}_t$), but they are in the same equivalence class in the sense that $\boldsymbol{\theta}'_t, \boldsymbol{\theta}_t$ should satisfy (8).

Also note that the learning of inverse mapping $\boldsymbol{f}'^{-1}_t$ in the encoder $q$ is *enforced* by consistency $(q(\mathbf{z}'|\boldsymbol{x}, y, t) = p(\mathbf{z}'|\boldsymbol{x}, y, t))$, we can just use an MLP for $\boldsymbol{r}'$ in the encoder, and we will have $\boldsymbol{r}'_t = \boldsymbol{f}'^{-1}_t$ if the MLP is flexible enough to contain $\boldsymbol{f}'^{-1}_t$. Similar situations prevail when identifiability is achieved by nonlinear ICA (Hyvärinen & Morioka, 2016; Hyvärinen et al., 2019; Khemakhem et al., 2020). $\qquad\square$

## 8.2 ON THE THREE IDENTIFICATION CONDITIONS

Ignorability given $\mathbf{z}$ means there is no correlation between factual assignment of treatment and counterfactual outcomes given $\mathbf{z}$, just as it is the case in RCT. Thus, it can be understood as *unconfoundedness* given $\mathbf{z}$, and $\mathbf{z}$ can be seen as the *confounder(s)* we want to control for. Positivity says the supports of $p(\mathbf{t} = t|\mathbf{x} = \boldsymbol{x}), t = 0, 1$ should be *overlapped*, and this ensures there are no impossible events in the conditions after adding $\mathbf{t} = t$, and the expectations can thus be estimated from observational data. Finally, consistent counterfactuals are well defined: given assignment of treatment $\mathbf{t} = t$, the *observational outcome* $\mathbf{y}$ should take the same value as the potential outcome $\mathbf{y}(t)$.

## 8.3 CONDITIONAL VAE

By adding a conditioning variable $\mathbf{c}$ (usually a class label), Conditional VAE (CVAE) (Sohn et al., 2015; Kingma et al., 2014) can give better reconstruction of observation of each class. The variational lower bound is

$$\log p(\mathbf{y}|\mathbf{c}) \geq \mathbb{E}_{\mathbf{z}\sim q} \log p(\mathbf{y}|\mathbf{z}, \mathbf{c}) - D_{\mathrm{KL}}(q(\mathbf{z}|\mathbf{y}, \mathbf{c})\|p(\mathbf{z}|\mathbf{c})) := \mathcal{L}_{CVAE}(\mathbf{y}, \mathbf{c}) \qquad (14)$$

The conditioning on $\mathbf{c}$ in the prior is usually omitted, since the dependence between $\mathbf{c}$ and the latent representation is also involved in the encoder $q$.

## 8.4 IDENTIFIABILITY OF REPRESENTATION IN SEC. 5.1 IS NOT ENOUGH

Consider how the *recovered* $\mathbf{z}'$ would be used. For a control group $(t = 0)$ data point $(\boldsymbol{x}, y, 0)$, the real challenge is to predict the counterfactual outcome $y(1)$. Taking the observation, the encoder will output a posterior sample point $\mathbf{z}'_0 = \boldsymbol{f}'^{-1}_0(y) = \mathcal{A}^{-1}_0(\mathbf{z}_0)$ (with zero outcome noise, the encoder degenerates to a delta function: $q(\mathbf{z}|\boldsymbol{x}, y, 0) = \delta(\mathbf{z} - \boldsymbol{f}'^{-1}_0(y)))$. Then, we should do *counterfactual inference*, using decoder with counterfactual assignment $t = 1$: $y'_1 = \boldsymbol{f}'_1(\mathbf{z}'_0) = \boldsymbol{f}_1 \circ \mathcal{A}_1(\mathcal{A}^{-1}_0(\mathbf{z}_0))$. This prediction can be arbitrary far from the truth $y(1) = \boldsymbol{f}_1(\mathbf{z}_0)$, due to the difference between $\mathcal{A}_1$ and $\mathcal{A}_0$. More concretely, this is because when learning the decoder, only the posterior sample of the treatment group $(t = 1)$ is fed to $\boldsymbol{f}'_1$, and the posterior sample is different to the true value by the affine transformation $\mathcal{A}_1$, while it is $\mathcal{A}_0$ for $\mathbf{z}'_0$.

## 8.5 TWO SPECIAL CASES OF BALANCING COVARIATE

**Definition 2** (Noiseless proxy). Random variable $\mathbf{x}$ is a noiseless proxy of random variable $\mathbf{z}$ if $\mathbf{z}$ is a function of $\mathbf{x}$ ($\mathbf{z} = \boldsymbol{\omega}(\mathbf{x})$).

Noiseless proxy is a special case of balancing covariate because if $\mathbf{x} = \boldsymbol{x}$ is given, we know $\mathbf{z} = \boldsymbol{\omega}(\boldsymbol{x})$ and $\boldsymbol{\omega}$ is a deterministic function, then $p(\mathbf{z}|\mathbf{x} = \boldsymbol{x}) = p(\mathbf{z}|\mathbf{x} = \boldsymbol{x}, \mathbf{t}) = \delta(\mathbf{z} - \boldsymbol{\omega}(\boldsymbol{x}))$. Also note that, a noiseless proxy always has higher dimensionality than $\mathbf{z}$, or at least the same.

Intuitively, if the value of $\mathbf{x}$ is given, there is no further uncertainty about $\mathbf{z}$, so the observation of $\boldsymbol{x}$ may work equally well to adjust for confounding. But, as we will see soon, a noiseless proxy of the true confounder does *not* satisfy positivity.

**Definition 3** (Injective proxy). Random variable $\mathbf{x}$ is an injective proxy of random variable $\mathbf{z}$ if $\mathbf{x}$ is an injective function of $\mathbf{z}$ ($\mathbf{x} = \boldsymbol{\chi}(\mathbf{z})$, $\boldsymbol{\chi}$ is injective).

Injective proxy is again a special case of noiseless proxy, since, by injectivity, $\mathbf{z} = \boldsymbol{\chi}^{-1}(\mathbf{x})$, i.e. $\mathbf{z}$ is also a function of $\mathbf{x}$.

Under this very special case, that is, if $\mathbf{x}$ is an injective proxy of the true confounder $\mathbf{z}$, we finally have $\mathbf{x}$ is a balancing score and satisfies strong ignorability, since $\mathbf{x}$ is a balancing covariate and a function of $\mathbf{z}$. To see this in another way, let $f = e \circ \boldsymbol{\chi}^{-1}$ and $\boldsymbol{b} = \boldsymbol{\chi}$ in Theorem 1, then $f(\mathbf{x}) = f(\boldsymbol{b}(\mathbf{z})) = e(\mathbf{z})$. By strong ignorability of $\mathbf{x}$, (2) has a simpler counterpart $\mu_t(\boldsymbol{x}) = \mathbb{E}(\mathbf{y}(t)|\mathbf{x} = \boldsymbol{x}) = \mathbb{E}(\mathbf{y}|\mathbf{x} = \boldsymbol{x}, \mathbf{t} = t)$. Thus, a regression of $\mathbf{y}$ on $(\mathbf{x}, \mathbf{t})$ will give a valid estimator of CATE and ATE.

However, a noiseless but *non*-injective proxy is *not* a balancing score, in particular, positivity might *not* hold. Here, a simple regression will not do. This is exactly because $\boldsymbol{\omega}$ is non-injective, hence multiple values of $\mathbf{x}$ that cause non-overlapped supports of $p(\mathbf{t} = t|\mathbf{x} = \boldsymbol{x}), t = 0, 1$ might be mapped to the same value of $\mathbf{z}$. An extreme example would be $\mathbf{t} = \mathbb{I}(\mathbf{x} > 0), \mathbf{z} = |\mathbf{x}|$. We can see $p(\mathbf{t} = t|\mathbf{x})$ are totally non-overlapped, but $\forall t, z \neq 0 : p(\mathbf{t} = t|\mathbf{z} = z) = 1/2$.

## 8.6 DETAILS AND ADDITIONAL RESULTS FOR EXPERIMENTS

### 8.6.1 SYNTHETIC DATA

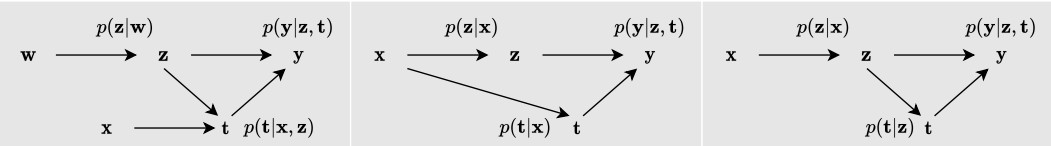

Figure 4: Graphical models for generating synthetic datasets. From left: IV, ignorability given $\mathbf{x}$, and non-balancing proxy $\mathbf{x}$. Note that in the latter two cases, reversing the arrow between $\mathbf{x}, \mathbf{z}$ does not change any independence relationships, and causal interpretations of the graphs remain the same.

Interestingly, linear outcome models seem harder for both methods, maybe because the two true linear outcome models for $t = 0, 1$ are more similar, and it is harder to distinguish and learning outcome models. Note that after generating the outcomes and before the data is used, we normalize the distribution of ATE of the 100 generating models, so the errors on linear and nonlinear settings are basically comparable.

You can find more plots for latent recovery at the end of the paper.

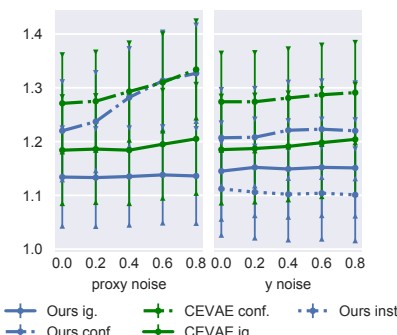

Figure 5: $\sqrt{\epsilon_{PEHE}}$ on linear synthetic dataset. Error bar on 100 random models. We adjust one of $\alpha, \beta$ at a time. Results for ATE and post-treatment are similar.

### 8.6.2 IHDP

IHDP is based on an RCT where each data point represents a child with 25 features about their birth and mothers. Race is introduced as a confounder by artificially removing all treated children with nonwhite mothers. There are 747 subjects left in the dataset. The outcome is synthesized by taking the covariates (features excluding race) as input, hence *ignorability* holds given the covariates. Following previous work, we split the dataset by 63:27:10 for training, validation, and testing.

### 8.6.3 POKEC

GCN takes the network matrix $\boldsymbol{G}$ and the *whole* covariates matrix $\boldsymbol{X} := (\boldsymbol{x}_1^T, \ldots, \boldsymbol{x}_M^T)^T$, where $M$ is user number, and outputs a representation matrix $\boldsymbol{R}$, again for all users. During training, we *select* the rows in $\boldsymbol{R}$ that correspond to users in training set. Then, treat this *training representation matrix* as if it is the covariates matrix for a non-networked dataset, that is, the downstream networks in conditional prior and encoder are the same as in the above two experiments, but take $(\boldsymbol{R}_{m,:})^T$ where $\boldsymbol{x}_m$ was expected as input. And we have respective selection operations for validation and testing.

We can still train CFVAE including GCN by Adam, simply setting the gradients of non-seleted rows of $\boldsymbol{R}$ to 0.

Note that GCN cannot be trained using mini-batch, instead, we perform batch gradient decent using full dataset for each iteration, with initial learning rate $10^{-2}$. We use dropout (Srivastava et al., 2014) with rate 0.1 to prevent overfitting.

The pre-treatment $\sqrt{\epsilon_{PEHE}}$ for `Age`, `District`, and `Join date` confounders are 1.085, 0.686, and 0.699 respectively, practically the same as the ATE errors.

### 8.7 ADDITIONAL PLOTS ON SYNTHETIC DATASETS

See next pages.

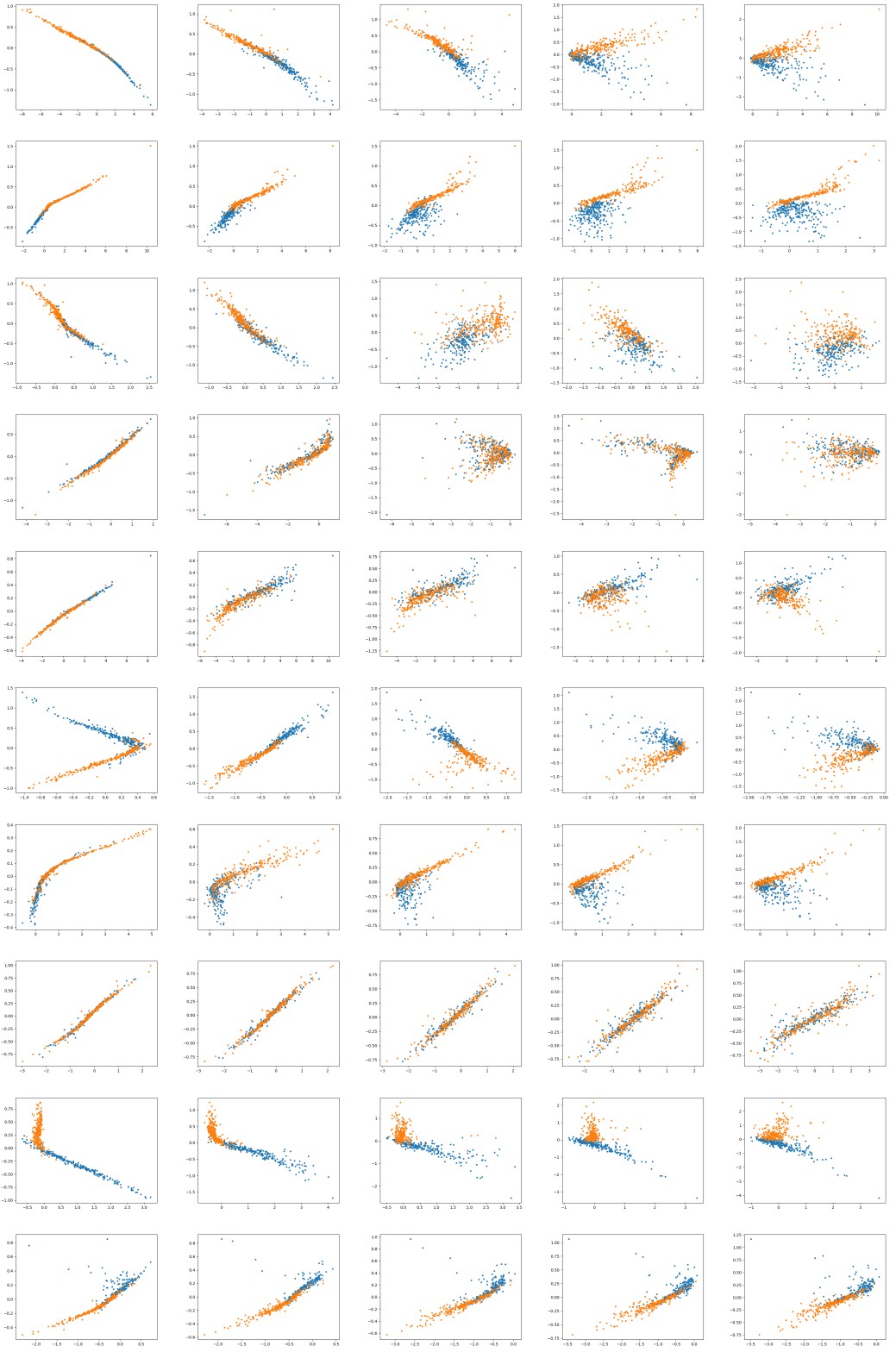

Figure 6: Plots of recovered-true latent under *unobserved confounding*. Rows: first 10 nonlinear random models, columns: *proxy* noise level.

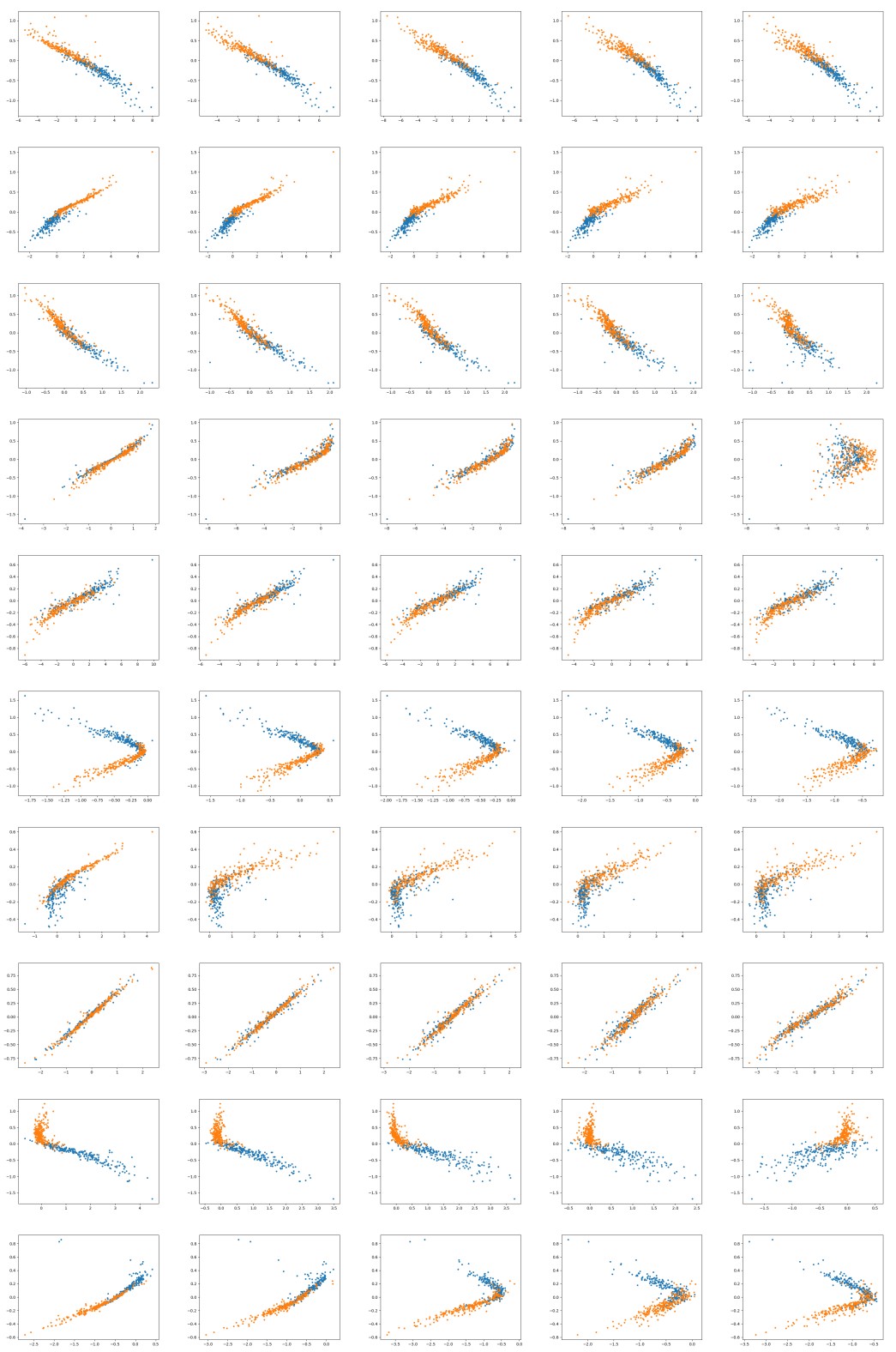

Figure 7: Plots of recovered-true latent under *unobserved confounding*. Rows: first 10 nonlinear random models, columns: *outcome* noise level.

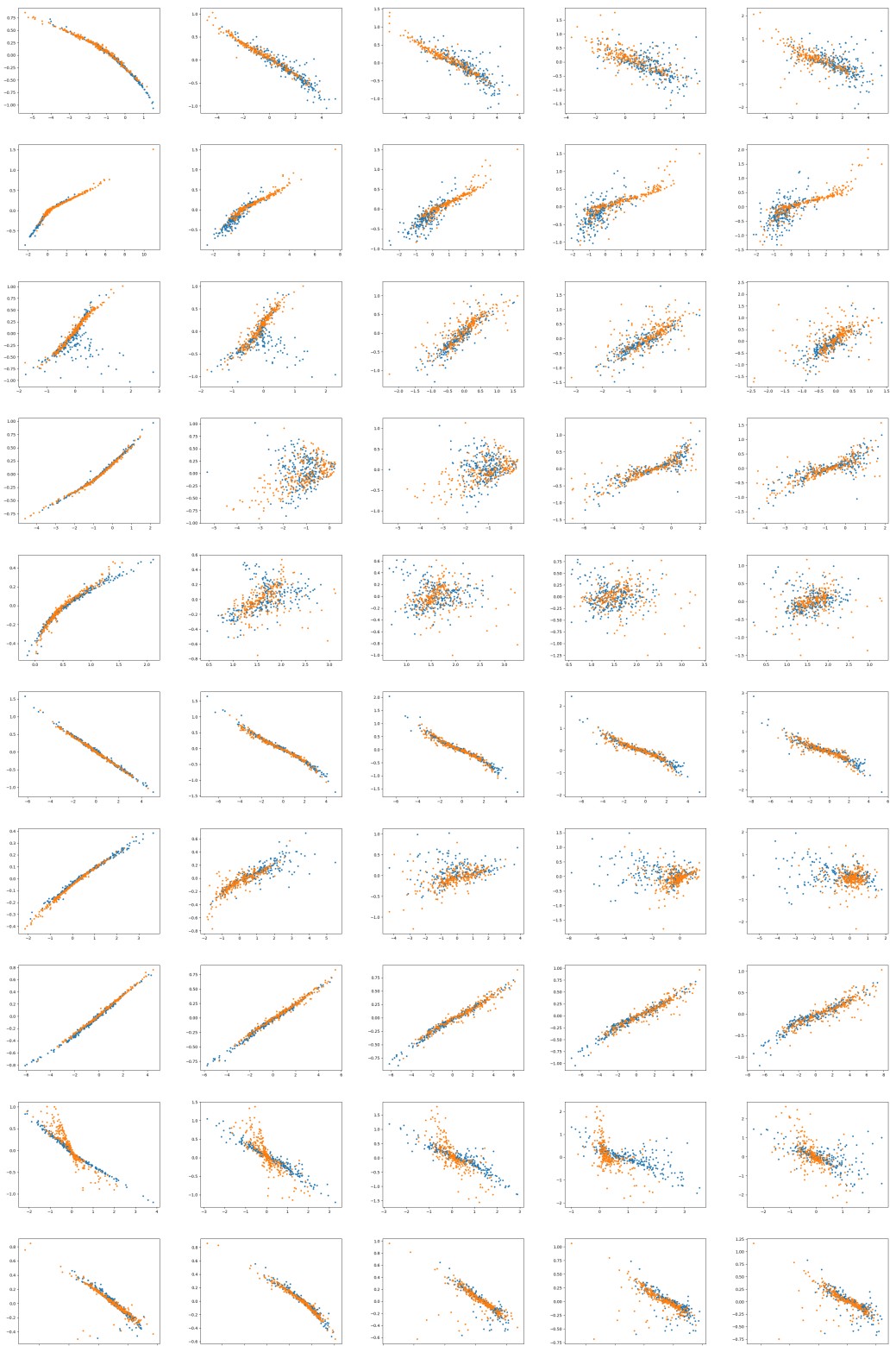

Figure 8: Plots of recovered-true latent when *ignorability* holds. Rows: first 10 nonlinear random models, columns: *proxy* noise level.

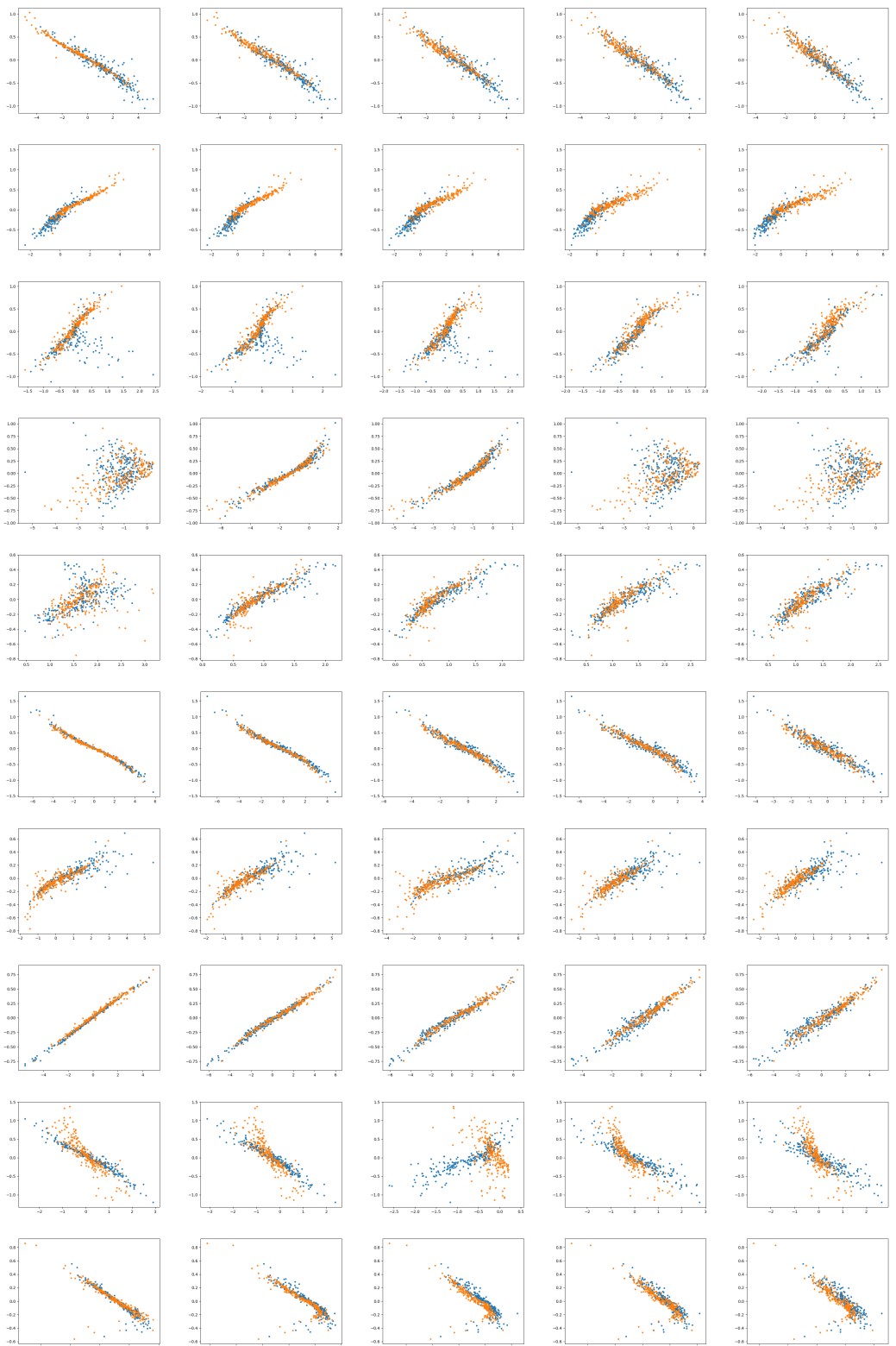

Figure 9: Plots of recovered-true latent when *ignorability* holds. Rows: first 10 nonlinear random models, columns: *outcome* noise level.

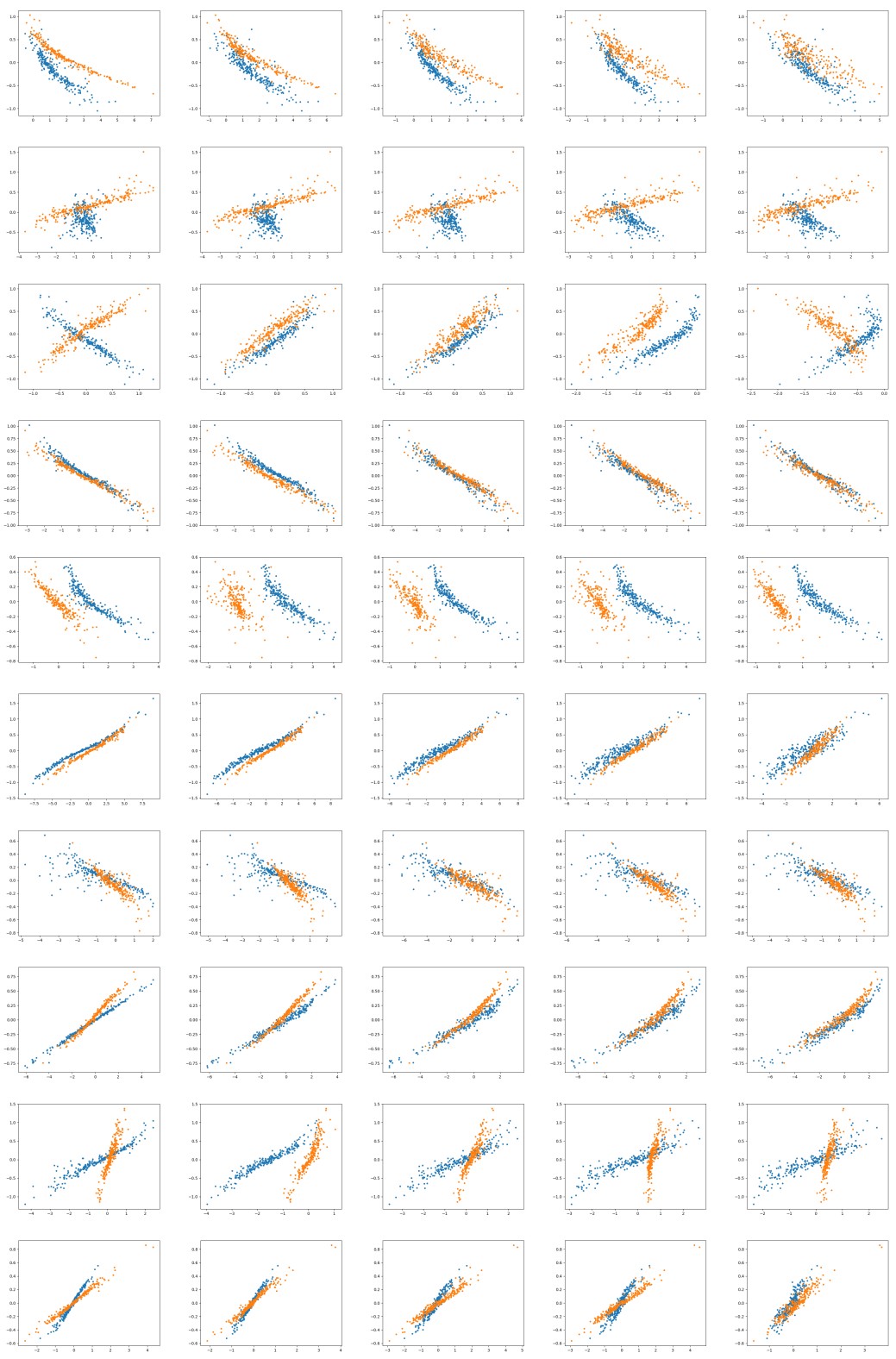

Figure 10: Plots of recovered-true latent when *ignorability* holds. Conditional prior *depends* on $t$. Rows: first 10 nonlinear random models, columns: *outcome* noise level. Compare to the previous figure, we can see the transformations for $t = 0, 1$ are *not* the same, confirming our Theorem 3.

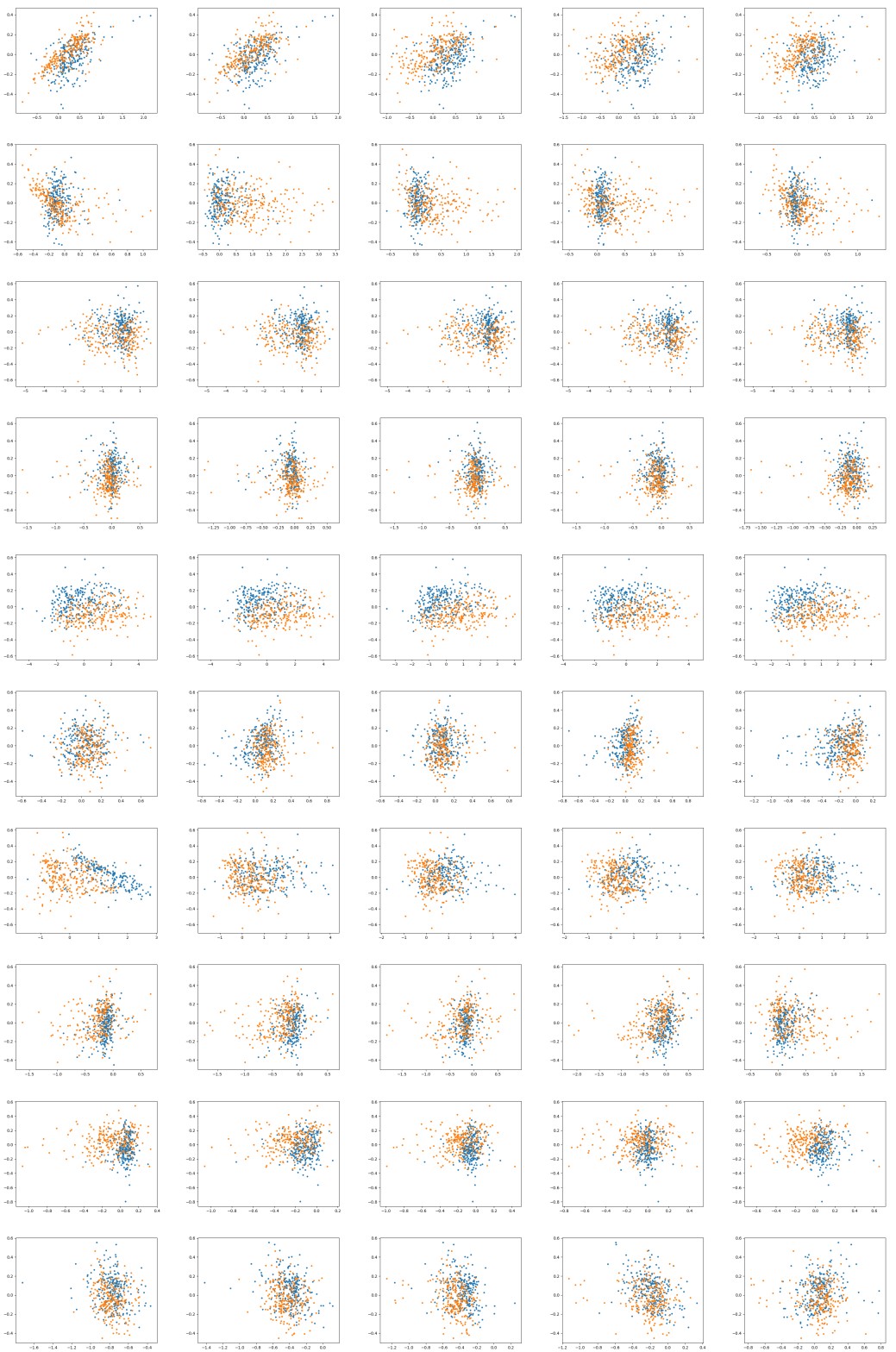

Figure 11: Plots of recovered-true latent on *IVs*. Rows: first 10 nonlinear random models, columns: *outcome* noise level.

