# OpenReview forum: "Identifying Treatment Effects under Unobserved Confounding by Causal Representation Learning"
_ICLR.cc/2021/Conference — Reject_

### Official Review · AnonReviewer4 · 2020-10-28
**Interesting attempt to address the difficult problem of identifiability in the context of non-linear latent confounding, but some points need further clarifications.**

**Rating:** 4
**Confidence:** 4

**Review:**

Summary:
The present paper introduces Counterfactual VAE (CFVAE), a generative learning method to estimate treatment effects under a latent unconfoundedness assumption. It builds on variational autoencoders (VAE) to learn causal representations. The authors provide identification results using recent results on nonlinear ICA (Khemakhem et al., 2020). They show that the confounder is identifiable up to an affine transformation.
The main contributions of this work are presented in Sections 5.1 and 5.2, where they derive identifiability of the treatment effect via identifiability of representation.
The theoretical claims are complemented with various synthetic and semi-synthetic experiments which also show that the proposed models can compete with and in certain cases improve upon state of the art.

Recommendation:
Reject. In summary, I am rather convinced that the contribution of this paper is important, but lacks clarity in its arguments and clarifications w.r.t. its positioning in the standard causal inference framework which it claims to make a new contribution to and its causal model/underlying assumptions about confounding are not stated clearly enough. I will read the rebuttal carefully and am willing to increase the score if the authors address the raised concerns.

Strong points:
 - This work provides a simplified and yet still practically useful framework that leverages recent nonlinear ICA result to provide an identification result in the causal inference framework.
 - The simulations, especially in Section 6.1 are well presented and commented.

Issues/Points that require clarification:
 - The positioning w.r.t. CEVAE and more generally w.r.t. to classical unconfoundedness assumptions is not very explicit in that the authors first state that their method requires weaker assumptions than CEVAE but from the identifiability discussion (Sec. 5.2) and the experiments (Sec. 6.1) it seems that the models are more different and not a weaker, resp. stronger version of each other. A simplified causal graph for CFVAE (maybe together with one for CEVAE) could make the comparison more explicit.
 - To my understanding, the balancing covariate assumption implies that $z$ is not directly related to $t$ (meaning in a causal graph there would be no arrow from $z$ to $t$) but only through $x$. So if one were to adjust for the treatment bias using only $x$, this should be sufficient to estimate the treatment effect, for instance by inverse propensity weighting. Thus again, could the authors provide a causal graph representing their assumptions about the role of each variable, most importantly $z$ and $x$, similar to Figure 1 of Louizos et al. (2017)?
 - The theory seems solid for the univariate case. How challenging would an extension to a multivariate case (i.e., multiple latent confounders) be? As a first step, have the authors looked at the empirical behaviour in case of multivariate confounders?
 - Related to the previous point, are there some concrete examples where the assumption of a latent univariate confounder is plausible/sufficient to capture all confounding?
 - The main claim of this work relies on results from Khemakhem et al. (2020), it would be helpful to have at least 1-2 sentences summarizing the main idea/result for identifiability of the iVAE in this cited article, so that the present work is more self-contained.
 - Concerning the pre-treatment prediction (p.6), have the authors verified empirically how their proposed alternative in the absence of post-treatment observation $y_t$ performs?
 - Would it be possible to add the parametric probabilistic PCA based approach by Mao et al. (2018) to the experiments? To my knowledge, this is the only work that gives a proved consistent estimator in the case of latent confounding. Also, this reference should be added in the related work section.

Minor comments (that did not impact the score):
 - p. 1: "effects of public policies or clinical trials" $>>$ replace clinical trials (clinical trial is a mean to estimate the effect of a new drug/treatment, not something that has an effect in itself)
 - p. 2: In many work $>>$ In many works
 - p. 2: if we apply $>>$ if we applied
 - p. 3: casual effects $>>$ causal effects
 - p. 3: CATE can be understood is an $>>$ CATE can be understood as an
 - p. 3: introduce/define $D_{KL}$ notation
 - p. 5: introduce/define $\delta$ notation
 - p. 6: check caption of Figure 2.
 - p. 7: descriptoins $>>$ descriptions

References:
 - Nathan Kallus, Xiaojie Mao, and Madeleine Udell. Causal inference with noisy and missing covariates via matrix factorization. In Advances in Neural Information Processing Systems, pp 6921–6932, 2020.
 - Ilyes Khemakhem, Diederik Kingma, Ricardo Monti, and Aapo Hyvarinen. Variational autoencoders and nonlinear ica: A unifying framework. In International Conference on Artificial Intelligence and Statistics, pp. 2207–2217, 2020.
 - Christos Louizos, Uri Shalit, Joris M Mooij, David Sontag, Richard Zemel, and Max Welling. Causal effect inference with deep latent-variable models. In Advances in Neural Information Processing Systems, pp. 6446–6456, 2017.

===================

Post Rebuttal Update:

While I think the proposal is interesting, I still think the proposed methodology lacks a clear presentation of the studied (causal inference) problem and statement of its underlying assumptions, as it has also been pointed out by other reviewers. Despite some clarifications from the authors I still vote for rejection as I believe the paper requires a major revision.

---

> ### Author Response · Authors · 2020-11-24
> **Clarifications**
>
> We reply the reviewer's concerns point by point:
>
> - (Comparison to CEVAE) We admit the statement "*Except this independence, we do not make any specific causal assumptions on the covariates*" in the introduction was a bit misleading and is **replaced** now. Compared to CEVAE, the main additional assumption of our method is the balancing covariate assumption, which is required to learn a affine transformation of the true confounder. However, CEVAE assumes directly that the true confounder can be recovered, and this, as implied by our Sec. 5, is arguably an even stronger assumption than balancing covariate. The graphical model (that needs *not* to have causal implications) for the decoder of CFVAE is presented in Figure 1, while the graphical model of our encoder is standard and thus omitted (see the caption of Figure 1). The encoder of CEVAE is nonstandard and more complex than ours. Please also see the reply to reviewer3 on our novelty compared to CEVAE and iVAE.
> - (Example satisfying our assumptions) For a proxy variable (e.g., Figure 1 of Louizos et al. (2017)) to further satisfy (t indep z|x), one possibility is noiseless proxy, defined in Appendix 8.5 (note that noiseless requirement violates causal faithfulness).
> - Our Theorem 2 (Theorem 3 in the updated version) in itself is not constrained to univariate latent z, but instead requires that the dimensionality of z is not larger than that of outcome y. So there might be a practical constraint because y is usually univariate. Experiments on multivariate artificial data will be added. Also see the following point.
> - (When univariate latent is sufficient) We can think in turns of the complexity of confounding. Imagine, if we have 10 independent latent confounders, and they have *linear* effects on both t and y. Then it is plausible that the total confounding effects can be modeled by single latent variable z, but complex enough functions relating t, y and z. On the other hand, as seen in 6.2 and 6.3, we can use multivariate z to achieve better performance even if y is univariate. Hence we think the practical usefulness of our method is promising.
> - We **updated** the paper, adding more on iVAE in 3.2.
> - Yes, we evaluated the pre-treatment performance (without observation y_t) in each experiment, and the results constantly matches that of post-treatment, or only slightly worse. In figure 3 6.1 and table 2 6.3, the reported results are pre-treatment. In table 1 6.2, we reported both pre/post-treatment results.
> - Thank you for the reference to Mao et al. (2018). It is indeed relevant. We **added** it in the related work. We will also add experimental comparison.
> - Thank you for the careful reading. We have **fixed** the typos and careless omissions.

---

### Official Review · AnonReviewer2 · 2020-10-28
**The paper makes some arbitrary assumptions**

**Rating:** 4
**Confidence:** 4

**Review:**

The paper proposes a new variant of VAE for estimating conditional treatment effects under unobserved confounding. It provides theoretical results on the identifiability of the confounder and the treatment effects under some assumptions. Experimental results are provided to demonstrate the performance of the proposed method.

Pros:

-The paper addresses an important problem

-The paper provides a theoretical analysis of the identifiability of representation and treatment effect.

Cons:

-I’m confused by the role of $x$ and its relation with the unobserved confounder $z$. In the works that do not consider unobserved confounders, I believe the covariates $x$ in the treatment effect $\mu_t(x)$ are the observed confounders and assumed to satisfy the ignorability condition. In this paper, are you assuming there exist no observed confounders? That would be a very unreasonable assumption.

-I think the ignorabiltiy assumption made before Eq. (2) is not correct. For (2) to hold, you’d need (y(0), y(1) independent t |z,x) to hold. I believe the propensity score should be p(t=1|z,x).

-The paper made several assumptions that look to me quite arbitrary, including (y indep x|z, t) and (t indep z|x). I’m not sure they are consistent with each other or with the ignorabiltiy assumption. What graphical models satisfy all these assumptions? The CFVAE structure in Figure 1 does not look like satisfying all these assumptions.

Overall, I vote for reject. I think the paper made some unreasonable assumptions.

Other comments:

-What is the graphical structure of the data-generating model in (10)?

-Why only compare with CEVAE on the synthetic datasets but not other methods?

---

> ### Author Response · Authors · 2020-11-24
> **Clarifications on assumptions**
>
> We reply the reviewer's concerns point by point:
>
> - The existence of observed confounders is allowed. The description of z as "unobserved confounders" in the paper was a bit confusing. Actually, since z is the latent variable(s) for VAE and is learned from covariates x by VAE, it can contain *all* confounders in principle. If there are observed confounders in x, our method will extract that part of x into z. We **updated** the paper to make this clear.
> - (Definitions of ignorability and propensity) We believe this point is also addressed by the above.
> - (Example satisfying our assumptions) A typical proxy variable x satisfies (y indep x|z, t) (e.g., Figure 1 of Louizos et al. (2017)), and for a proxy variable to further satisfy (t indep z|x), one possibility is noiseless proxy, defined in Appendix 8.5 (note that noiseless requirement violates causal faithfulness). On the other hand, our Fig. 1 shows the graphical model, that needs *not* to have causal implications, of the decoder of CFVAE (which is built to easily satisfy the assumptions of iVAE).
> - Graphical models for generating synthetic datasets are now **added** in Appendix. Note that eq (10) is a general equation which will be used to generate 3 different causal settings, as described in the paragraph "We experiment on three different causal settings...".
> - More comparisons on synthetic datasets will be added.

---

### Official Review · AnonReviewer3 · 2020-10-29
**Theoretical motivation for VAE-based CATE estimation under confounding with proxy variables.**

**Rating:** 6
**Confidence:** 2

**Review:**

Summary:

This paper provides a method for using a VAE with proxy variables to estimate CATE in a model with latent confounding by recovering a conditional distribution over the latent confounders. Building upon results from Khemakhem et.al. 2020, the confounding can be identified if the latent variable is parameterized by an exponential family distribution dependent on the proxy variable, and the outcome variable is an injective function of the latents with (small) additive noise. Conceptually, the paper is similar in goal to Louizos et al. 2017, with the main difference seeming to be a stronger theoretical base.

Reasons for score:

Overall, I would rate this paper as a borderline accept. The work seems to provide a strong theoretical background for VAE-based models to be used for identification purposes with latent confounding when proxy variables are present, allowing to explicitly specify conditions under which approaches modelling confounders for adjustment can work. I find this an interesting result that validates the intuitions behind previous works. My main concern with the paper is an unclear comparison to Louizos et. al, as well as an unclear specification of the necessary assumptions, which only becomes clear after reading Khemakhem et. al. Finally, given that this work seems to be a direct combination of these two previous works, I have some lingering doubts about the result's novelty, which would be greatly helped if the paper included a more detailed comparison.

Pros:

- As mentioned, provides a strong theoretical backing for a VAE-based approach to CATE, giving a clear set of conditions under which such approaches can be used
- Provides convincing experimental evidence that the proposed method, CFVAE, compares favorably with existing approaches.
- Very clearly lays out the relevant background literature, allowing to cleanly place this work in context.

Cons:

- The difference between CEVAE and CFVAE should be clarified. For example, the statement "CEVAE assumes a specific causal graph where the covariates should be independent of the treatment given the confounder." seems to contradict the end of page 3 in Louizos et. al., where it looks like they state that there can be a direct edge from X to t. Could the authors comment on this?
- While the fact that the paper was using results from Khemakhem et. al. was made very clear, I needed to go to the original paper for a clear specification of the precise conditions required for the method to work (exponential family, outcome is function of latent with small additive noise, etc). It would be useful if these conditions were listed clearly and centrally in the paper, without assuming familiarity with this previous work.

Comments:

I think it might be useful to include CEVAE in Fig 1, and add it to the discussion in section 3.2/4. In general, showing the precise difference between the two VAE formulations would go a long way towards making the contribution's novelty easier to understand.

---

> ### Author Response · Authors · 2020-11-24
> **Comparisons and Novelties**
>
> We reply the reviewer's concerns as following:
>
> ### Comparison with CEVAE
>
> - The similarity to CEVAE is mainly on the initial motivation by proxy variable and the employment of VAE.
> - CFVAE is quite different to CEVAE in *architecture*, both its decoder and (simpler) encoder. Please see Figure 1 and its description.
> - CEVAE assumes implicitly p(x,t|z)=p(x|z)p(t|z) (x \indep t |z) in their eq (2) for building the decoder. This means direct edge from X to t is actually not allowed, unless causal faithfulness is violated.
> - We will add more detailed comparison (with figures), possibly in Appendix.
>
> ### Assumptions from iVAE
>
> - The beginning of Sec. 5.1 introduced the assumptions under our own setting, though in a less formal way. We **updated** the paper, presenting a formal theorem, and also saying more about iVAE in 3.2.
>
> ### More comments on novelty
>
> - CEVAE is tailored and limited to the setting of standard proxy variable (as in their Fig. 1), but CFVAE works under more general settings, given only the independence is satisfied.
> - On the other hand, the derivation of the variational lower bound of CFVAE from (5) to (6) is more principally built on identifiability of VAE and identification equation (2).
> - iVAE did not make any connection to causal inference. In particular, first, we motivate the independence of iVAE by that of proxy, and second, there is no place of the treatment variable in iVAE and thus it is not directly applicable to the estimation of treatment effect, and we add the conditioning on treatment to address this.

---

### Official Review · AnonReviewer1 · 2020-10-30
**Arguments about identification need to be more careful; Key assumption makes the problem trivial**

**Rating:** 3
**Confidence:** 4

**Review:**

3342 Identifying Treatment Effects Under Unobserved Confounding
# Summary

The authors propose a representation learning method for estimating causal effects in the presence of unobserved confounding when covariates that act as proxies for a latent confounder are available. The authors connect the problem to some recent results on the partial identifiability of VAE and non-linear ICA models. The authors lay out some conditions under which the causal effect may be identifiable and propose a VAE-based model, CFVAE, that enforces some of these conditions on the estimated model. They compare CFVAE to a previously proposed VAE algorithm, CEVAE, and other methods that are designed to work under unconfoundedness.

# Feedback

Identification of causal effects in the presence of proxy variables for unobserved confounders is an important but subtle problem. However, this means that the bar for making contributions in this area needs to be high. Because the conditions for identification can’t be falsified in a particular application, making unclear statements about when the effects of interest are and are not identifiable is very important. Readers who misunderstand will only experience silent failures and make poor decisions as a result.

Unfortunately, I don’t believe this paper meets this bar of clarity. I think that the authors explore some interesting connections to recent work on identifiability in latent variable models, and understanding what these results imply for causal inference is important. In particular, the observation that one might estimate a decoder up to a _different_ affine transformation in the treated and control arms seems like a useful insight. But the results in the paper are incomplete, disorganized, and in some cases wrong. I’ll list out a few general points here, then discuss some substantive issues with the paper’s specific argument.

## General points about identifiability arguments

### Identifiability is not a property of the method

Identifiability is a property of the _data generating process_ and not a property of the model being used to do estimation. If the causal effect of interest is not identifiable in the process that generated the data, then using an “identifiable model” to estimate the causal effect will not solve the problem.

The exposition in the paper seems to argue that using the right model will make the causal effect identifiable. This may just be a matter of unclear writing, but this is a broader point of confusion in the ML community, so it’s important that the authors be clear on this point.

Here, I think it would make sense for the authors to state what their assumptions are about the data generating process, separately from the parameterization of their model. Reading the paper, the distinction between these two layers of assumptions was unclear.

### Relaxing identifying assumptions is not an option

In the same vein, if there are assumptions that the authors need to make to eliminate identification failure modes, then showing that the model “works” when those assumptions are relaxed does not inspire confidence. Unlike standard model misspecification which can be detected and debugged based on observables, making the wrong identifying assumptions in a model can result in silent failure, where the model can fit the observed data perfectly, but return a causal effect estimate that converges to the wrong place, or doesn’t converge at all.

If one is able to relax the identifying assumptions and still see success in experiments, this means either (a) the assumptions were unnecessary, or (b) the experiments did not probe the method well enough.

### Assumptions needs to be stated clearly, with implications clearly highlighted

When making identification arguments, assumptions play the role of eliminating equally plausible causal explanations of the observed data, until only the true one can remain (if the assumptions are true). These are not the kinds of assumptions that eliminate exotic corner cases; instead, they eliminate cases like the most obvious explanation for the observed data like the absence of unobserved confounding. Bounding arguments like the Manski and Kallus et al papers that are cited in the introduction construct the full set of causal explanations that identifying assumptions must narrow down to a point.

All of this is to say that clearly stating assumptions, and the cases they eliminate, is essential for any identification argument. In the paper as it is written now, many assumptions are made implicitly or in passing, and it is unclear which assumptions are made for illustration (e.g., the noise on the outcome going to zero) and which assumptions are essential for the argument in general. The assumptions are always framed as “mild” and do not highlight situations that the assumptions eliminate (i.e., in which cases they would fail to hold).

## Specific Concerns

### Balancing covariate implies that the naive estimator “just works”

The primary identifiability result of the paper involves the assumption that the observed covariates are “balancing covariates”, satisfying t \indep z | x. The authors argue that this is a weaker condition than requiring that x satisfy unconfoundedness. This may be true, but the gap between the two assumptions is, at most, a set of knife-edge violations of faithfulness. In terms of estimating causal effects, simply doing the standard covariate with x and ignoring z would give the right answer.

This can be shown in two ways. First, graphically, t \indep z | x implies that there is no backdoor path through z from t to y when you condition on z. So z doesn’t induce any non-causal association between y and t. Secondly, using the standard adjustment formula:

\mu_t(x) = E[ E[Y | X = x, T = t, Z = z] ]
= \int_z E[Y | X = x, T = t, Z = z] p(z | x) dz
= \int_z E[Y | X = x, T = t, Z = z] p(z | x, t) dz  (using the balancing covariate property)
= E[Y | X = x, T = t]

In particular, the naive regression function E[Y |  X = x, T = t] only fails in cases where p(z | x, t) \neq p(z | x); i.e., when the distribution of the latent variable is different in the two observed treatment arms even after conditioning on x. The balancing covariate property eliminates this possibility.

This also means that the VAE model specified can only generate data where the latent variable z does not introduce confounding.

## Other Concerns

 * The adjustment formula in equation (2) is wrong. The last integral should be with respect to p(z | x), not p(z | x, t) (see the argument above).

 * There is a substantive difference between estimating individual level causal effects given the observed outcome (a counterfactual query) versus estimating the CATE. The authors do divide this into “pre-treatment” and “post-treatment prediction”, but the counterfactual query presents additional identification questions. In particular, whether the reported expectation is correct depends on Cov(y(1), y(0) | z, t), which is never observable. None of the identifying assumptions in the paper make any arguments about this quantity, so the models in the paper are making implicit strong assumptions here.

 * The f^{-1}(y) notation in the paper is very unclear in the case that there is actually outcome noise for y. The function f relates the latent z to the expectation of y, not y itself. When y includes independent noise, the distribution of f^{-1}(y) does not yield the marginal distribution of z; you need to deconvolve the independent noise in y, which is non-trivial.

---

> ### Author Response · Authors · 2020-11-24
> **Clarity improved and contributions emphasized**
>
> We thank the constructive comments, and reply the reviewer's concerns as following:
>
> General points:
>
> - We totally agree that "*Identifiability is a property of the data generating process*". In this work, we made a *parametric* assumption, that the true conditional prior distribution p(z|x,t) should be in an exponential family. Some readers would be confused and think using exponential family conditional prior "will make the causal effect identifiable".  We **updated** the paper to make this clearer, rewording Theorem 2 (now Theorem 3) and adding explanations above it.
> - (Non-parametric identification?) We are not sure if the reviewer were suggesting us to first present conditions for *non-parametric* identification. We plan to leave this to future work, by analysis of our assumptions in light of, for example, (Kuroki and Pearl 2014) and (Miao et al 2018). We would be grateful if the reviewer could give further suggestions on this.
> - (Reviewer's point 2 and 3 replied together) Given the experimental results, we tend to believe some assumptions we made are not necessary. And we will follow the reviewer's suggestion, examine and highlight which assumptions are really necessary to eliminate certain causal situations, and which assumptions are "non-causal" and could thus be relaxed. As a first step, we **updated** the paper, listing our assumptions more clearly.
>
> Specific concerns:
>
> - The reviewer's second argument (by adjustment eq.) requires that p(t|x) should be *positive* at any points. This opens the possibility that ignorability might also fail when positivity does not hold. We would like to emphasize that our method works *without* positivity given x, but naïve regression does not. Moreover, even if positivity given x is satisfied, naïve regression still has practical issues: given finite data and the high dimensionality of x, it is quite possible that for many values of x, we do not actually have data for both t. But since our method learns a lower dimensional z, this problem will be largely addressed.
>
> Other concerns:
>
> - (Rightmost side of eq (2)) Sorry, this was a typo, and we **fixed** it.
> - (Individual causal effects?) We are not sure if we understand this concern correctly, but we reply as following. By "individual level" causal effects, we mean the CATE is conditioned on a high dimensional and highly diverse set of covariates x, such that *in practice* an individual can be identified by x. We do *not* intend to tackle the problem of identifying individual causal effects in the *proper* sense. Note also that, under 0 noise limit, given (z, t), y_t=f(z, t) is deterministic.
> - (Noisy y and f^{-1}(y)) We agree the outcome noise will complex the problem and in particular the recovered z' will not be a deterministic function of true z (as shown in 6.1), but the f^{-1}(y) notation itself is still meaningful even if y is noisy.

---

### Author Response · Authors · 2020-11-24
**Paper updated**

We thank the reviewers for their time and thoughtful comments. We have updated the paper, and the major changes that would be interesting to all are as following:

1. Presenting the identifiability of our model in a formal theorem (now Theorem 2).
2. Rewording Theorem 2 (now Theorem 3) and adding explanations above it.
3. Listing our assumptions more clearly.

---

### Comment · ~Pengzhou_Abel_Wu1 · 2021-02-18
**Post-decision update on Arxiv**

## Edit@Jan 2022: Although we had tried our best, the revision mentioned below has a *flawed* theoretical formulation. Please *avoid* it and refer to the new papers mentioned above!



Thanks to the thoughtful reviews here, we made a major update of our work. We believe most, if not all, issues raised here are addressed in the revision. About ***80%*** of the paper is rewritten, and ***theoretical part is totally new***. We do not post here the link to the new version, but it can be easily found on my google scholar page.

Some highlights are given below.

1. The balancing covariate assumption is dropped. Instead, we base our theory on general concepts of sufficient scores, specifically, *Bt-score* (Definition 2 and Theorem 1) that is new to machine learning methods and arguably more applicable than balancing score.
2. The zero (small) outcome noise assumption is no longer needed. Instead, we build on general additive noise model, and, particularly, *injective outcome model* (Lemma 1).
3. Assumptions on identifiable generating process, together with examples, are given, separated from model specifications.
4. Based on the above, we give a new, clearer and much more general identification result (Theorem 2).
5. We give consistent estimator by VAE in Sec 4.3, separated from identification. Particularly, we highlight our *balanced* estimator under confounding (Theorem 3), based on identifiability of VAE.

Other important issues addressed:

1. A causal graph of our problem setting (Figure 1) is given.
2. Stress the fundamental difficulty of unobserved confounding in Introduction and Sec. 2.
3. Thorough comparisons to CEVAE, and a formal statement with brief proof of our model identifiability based on iVAE, are given in Appendix.
4. Thorough rewrite of the interpretations of experiments, based on the new theory.

---

### Comment · ~Pengzhou_Abel_Wu1 · 2021-10-07
**Updates**

https://arxiv.org/abs/2109.15062 contains some newest ideas (e.g., mean independence and moment restriction) of the series of work starting from here, while https://arxiv.org/abs/2110.05225 deals with limited overlap and is by far the most polished!

---

### Comment · ~Pengzhou_Abel_Wu1 · 2022-01-30
**Accepted to ICLR 2022**

A paper originating from here got accepted to ICLR 2022! We are glad that the meta-review acknowledges the paper deals with an “*important and understudied*” problem.

https://openreview.net/forum?id=q7n2RngwOM

---

### Decision · Program_Chairs · 2021-01-07
**Final Decision**

**Decision:**

Reject

**Comment:**

The reviewers noted that this is an important, interesting but difficult topic. They appreciated that the authors clarified their assumptions in the theorem statements. Nevertheless, they recommend the authors to detail in depth when the method work better than the method where only the covariates are adjusted. They still think that the paper would require major modifications to be considered for publication hence the decision is rejection the paper.